# CROSS-DOMAIN GRAPH DATA SCALING: A SHOWCASE WITH DIFFUSION MODELS

## ABSTRACT

Models for natural language and images benefit from data scaling behavior: the more data fed into the model, the better they perform. This 'better with more' phenomenon enables the effectiveness of large-scale pre-training on vast amounts of data. However, current graph pre-training methods struggle to scale up data due to heterogeneity across graphs. To achieve effective data scaling, we aim to develop a general model that is able to capture diverse data patterns of graphs and can be utilized to adaptively help the downstream tasks. To this end, we propose *UniAug*, a universal graph structure augmentor built on a diffusion model. We first pre-train a discrete diffusion model on thousands of graphs across domains to learn the graph structural patterns. In the downstream phase, we provide adaptive enhancement by conducting graph structure augmentation with the help of the pre-trained diffusion model via guided generation. By leveraging the pre-trained diffusion model for structure augmentation, we consistently achieve performance improvements across various downstream tasks in a plug-and-play manner. To the best of our knowledge, this study represents the first demonstration of a data-scaling graph structure augmentor on graphs across domains.

## 1 INTRODUCTION

The effectiveness of existing foundation models (Radford et al., 2021; Touvron et al., 2023; Kirillov et al., 2023) heavily relies on the availability of substantial amounts of data, where the relationship manifests as a scaling behavior between model performance and data scale (Kaplan et al., 2020). Consistent performance gain has been observed with the increasing scale of pre-training data in both Natural Language Processing (Kaplan et al., 2020; Hoffmann et al., 2022) and Computer Vision (Abnar et al., 2022; Zhai et al., 2022) domains. This data scaling phenomenon facilitates the development of general models endowed with extensive knowledge and effective data pattern recognition capabilities. In downstream applications, these models are capable of adaptively achieving performance gains across different tasks.

In the context of graphs, the availability of large-scale graph databases (Rossi & Ahmed, 2015; Hu et al., 2020; Leskovec & Krevl, 2014) enables possible data scaling across datasets and domains. Existing works have demonstrated graph data scaling following two limited settings: in-domain pre-training (Xia et al., 2023; Liu et al., 2024c) and task-specific selection for pre-training data (Cao et al., 2023). During the pre-training process, each graph in the pre-training pool must be validated as in-domain or relevant to the downstream dataset. Given a specific domain or task, the crucial discriminative data patterns are likely confined to a fixed set (Mao et al., 2024c), leaving other potential patterns in diverse graph data distribution as noisy input. In terms of structure, graphs from different domains are particularly composed of varied patterns (Milo et al., 2002), making it hard to transfer across domains. For example, considering the building blocks of the graphs, the motifs shared by the World Wide Web hyperlinks only partially align with those shared by genetic networks (Milo et al., 2002). Therefore, closely aligning the characteristics of the pre-training graphs and the downstream data both in feature and structure is essential for facilitating positive transfer (Cao et al., 2023). As a consequence, the necessity of such meticulous data filtering restricts these methods from scaling up graphs effectively, as they can only utilize a small part of the available data. Given the limitation of the graph pre-training methods, a pertinent question emerges: *How can we effectively leverage the increasing scale of graph data across domains?*

Rather than focusing solely on data patterns specific to particular domains, we aim to develop a model that has a comprehensive understanding of data patterns inherent across various types of graphs. In line with the principles of data scaling, we hypothesize that incorporating a broader range of training datasets can help the model build an effective and universal graph pattern library, avoiding an overemphasis on major data patterns specific to any single dataset (Mao et al., 2024a). To construct such a general-purpose model, we propose to utilize a diffusion model operating only on the structure as the backbone, for the following key reasons. (1) Unlike features, graph structures follow a uniform construction principle, namely, the connections between nodes. This allows for positive transfer across domains when the upstream and downstream data exhibit similar topological patterns (Cao et al., 2023). In particular, while the graph representations of neurons and forward electronic circuits are derived from distinct domains, they still share common motifs (Milo et al., 2002). (2) Current supervised and self-supervised methods tend to capture only specific patterns of graph data, with models designed for particular inductive biases (Mao et al., 2024a;c; Xu et al., 2018). For instance, graph convolutional networks (GCNs) excel in node-level representation learning by emphasizing homophily, whereas graph-level representation learning benefits from expressive GNNs capable of distinguishing complex graph structures. (3) We opt for a structure-only model due to the heterogeneous feature spaces across graphs, which often include missing features or mismatched semantics (Mao et al., 2024b). For instance, node features yield completely different interpretations in citation networks, where they represent keywords of documents, compared to molecular networks, where they denote properties of atoms. To this end, we pre-train a structure-only diffusion model on thousands of graphs, which serves as the upstream component of our framework.

In the downstream stage, we employ the pre-trained diffusion model as a **Uni**versal graph structure **Aug**mentor (**UniAug**) to enhance the dataset, where diffusion guidance (Ho & Salimans, 2022; Dhariwal & Nichol, 2021; Gruver et al., 2024) is employed to align the generated structure with the downstream requirements. Specifically, we generate synthetic structures with various guidance objectives, and the resulting graphs consist of *generated structures* and *original node features*. This data augmentation paradigm strategically circumvents feature heterogeneity and fully utilizes downstream inductive biases by applying carefully designed downstream models to the augmented graphs in a plug-and-play manner. Empirically, we apply *UniAug* to graphs from diverse domains and consistently observe performance improvement in node classification, link prediction, and graph property prediction. To the best of our knowledge, this study represents the first demonstration of a data-scaling graph structure augmentor on graphs across domains.

## 2 PRELIMINARY AND RELATED WORK

**Learning from unlabeled graphs** Graph self-supervised learning (SSL) methods provide examples of pre-training and fine-tuning paradigm (Hu et al., 2019; Hou et al., 2022; Kim et al., 2022; You et al., 2021; Xu et al., 2021). However, these methods benefit from limited data scaling due to feature heterogeneity, structural pattern differences across domains, and varying downstream inductive biases. It is worth mentioning that DCT (Liu et al., 2024a) presents a pre-training and then data augmentation pipeline on molecules. Despite its impressive performance improvement on graph-level tasks, DCT is bounded with molecules and thus the use cases are limited.

**Graph data augmentation** There have been many published works exploring graph data augmentation (GDA) since the introduction of graph neural networks (GNNs), with a focus on node-level (Park et al., 2021; Liu et al., 2022b; Azabou et al., 2023), link-level (Zhao et al., 2022; Nguyen & Fang, 2024), and graph-level (Han et al., 2022; Ling et al., 2023; Luo et al., 2022; Liu et al., 2022a; Kong et al., 2022). These GDA methods have been generally designed for specific tasks or particular aspects of graph data. In addition, they are often tailored for a single dataset and struggle to transfer to unseen patterns, which limits their generalizability to a broader class of applications.

**Diffusion models on graphs** Diffusion models (Ho et al., 2020; Song et al., 2021; Rombach et al., 2022) are latent variable models that learn data distribution by gradually adding noise into the data and then recovering the clean input. Existing diffusion models on graphs can be classified into two main categories depending on the type of noise injected, i.e. Gaussian or discrete. Previous works employed Gaussian diffusion models both on general graphs (Niu et al., 2020; Jo et al., 2022) and molecules (Shi et al., 2021; Xu et al., 2022). However, adding Gaussian noise into the adjacency matrix will destroy the sparsity of the graph, which hinders the scalability of the diffusion models (Haefeli et al., 2022).

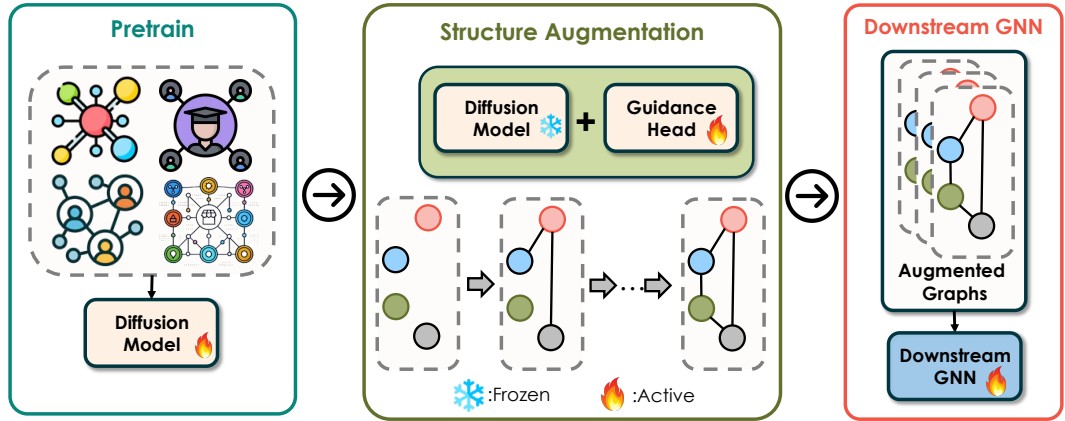

Figure 1: The pipeline of *UniAug*. We pre-train a diffusion model across domains and perform structure augmentation on the downstream graphs. The augmented graphs consist of *generated structures* and *original node features* and are then processed by a downstream GNN.

Recent works adapted discrete diffusion models to graphs with categorical transition kernels (Vignac et al., 2023; Chen et al., 2023b;a). We denote the adjacency matrix of a graph as $\mathbf{A}^0 \in \{0,1\}^{n \times n}$ with $n$ nodes. With details in Appendix A, we write the *forward process* to corrupt the adjacency matrix into a sequence of latent variables as Bernoulli distribution

$$q\left(\mathbf{A}^t \mid \mathbf{A}^{t-1}\right) = \text{Bernoulli}\left(\mathbf{A}^t; \alpha^t \mathbf{A}^{t-1} + \left(1 - \alpha^t\right) \pi\right),$$

$$q\left(\mathbf{A}^{t-1} \mid \mathbf{A}^t, \mathbf{A}^0\right) = \frac{q\left(\mathbf{A}^t \mid \mathbf{A}^{t-1}\right) q\left(\mathbf{A}^{t-1} \mid \mathbf{A}^0\right)}{q\left(\mathbf{A}^t \mid \mathbf{A}^0\right)}, \tag{1}$$

where $\pi$ is the converging non-zero probability, $\alpha^t$ is the noise scale, and $\bar{\alpha}^t = \prod_{i=1}^{t} \alpha^i$. Under predict-$\mathbf{A}^0$ parameterization, the *reverse process* denoise the adjacency matrix with a Markov chain

$$p_\theta\left(\mathbf{A}^{t-1} \mid \mathbf{A}^t\right) \propto \sum_{\widetilde{\mathbf{A}}_0} q\left(\mathbf{A}^{t-1} \mid \mathbf{A}^t, \widetilde{\mathbf{A}}_0\right) \tilde{p}_\theta\left(\widetilde{\mathbf{A}}_0 \mid \mathbf{A}^t\right), \tag{2}$$

where $\tilde{p}_\theta(\widetilde{\mathbf{A}}_0 \mid \mathbf{A}^t)$ represents the denoising network that predicts the original adjacency matrix from the noisy adjacency matrix. The parameters are estimated by optimizing the variational lower bound on the negative log-likelihood (Austin et al., 2021)

$$L_{\text{vb}} = \sum_{t=2}^{T} \mathbb{E}_{q(\mathbf{A}^t \mid \mathbf{A}^0)} \left[ D_{\text{KL}}\left(q\left(\mathbf{A}^{t-1} \mid \mathbf{A}^t, \mathbf{A}^0\right) \| p_\theta\left(\mathbf{A}^{t-1} \mid \mathbf{A}^t\right)\right) \right]$$
$$- \mathbb{E}_{q(\mathbf{A}^1 \mid \mathbf{A}^0)} \left[\log p_\theta\left(\mathbf{A}^0 \mid \mathbf{A}^1\right)\right] + \mathbb{E}_{q(\mathbf{A}^0)} \left[ D_{\text{KL}}\left(q\left(\mathbf{A}^t \mid \mathbf{A}^0\right) \| p\left(\mathbf{A}^t\right)\right) \right]. \tag{3}$$

## 3 METHOD

In this section, our goal is to build *UniAug* to understand the diverse structure patterns of graphs and perform data augmentation with a range of objectives. As illustrated in Fig.1, *UniAug* consists of two main components: a pre-trained diffusion model and the downstream adaptation through structure augmentation. We first collect thousands of graphs from varied domains with diverse patterns. To construct a general model free of downstream inductive biases, we train a self-conditioned discrete diffusion model on graph structures. In the downstream stage, we train an *MLP guidance head* on top of the diffusion model with objectives across different levels of granularity. We then augment the downstream dataset by generating synthetic structures through guided generation, where the augmented graph is composed of *generated structures* and *original node features*. Subsequently, we apply the augmented data to train a task-specific model for performing downstream tasks. Below, we elaborate on the data collection process, the architecture of the discrete diffusion model, and the guidance objectives employed.

### 3.1 PRE-TRAINING DATA COLLECTION

In light of the data scaling spirit, we expect our pre-training data to contain diverse data patterns with sufficient volume. As graphs from different domains exhibit different patterns (Milo et al., 2002), we wish to build a collection of graphs from numerous domains to enable a universal graph pattern library with pre-training. Within the publicly available graph databases, Network Repository (Rossi & Ahmed, 2015) provides a comprehensive collection of graphs with varied scales from different domains, such as biological networks, chemical networks, social networks, and many more. Among the thousands of graphs in the Network Repository, some of them contain irregular patterns, including multiple levels of edges, extremely high density, et cetera. To ensure the quality of the

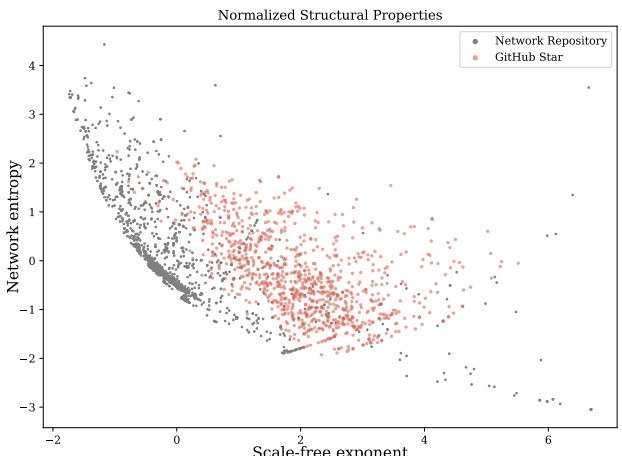

Figure 2: Normalized structural properties of Network Repository and Github Star. We enlarge the distribution coverage of our graph collection by combining both datasets.

graphs, we analyze the graph properties following Xu et al. (Xu et al., 2023) and filter out the outliers. In addition, we observe that the coverage of graphs in the Network Repository is incomplete according to the network entropy and scale-free exponent, as we observe a relatively scattered space in the middle of Fig. 2. To fill in the gap, we include a subset of the GitHub Star dataset (Rozemberczki et al., 2020) by random sampling 1000 graphs into our graph collection. The selected graphs are utilized to train a discrete diffusion model.

## 3.2 PRE-TRAINING THROUGH DIFFUSION MODEL

Diffusion models have demonstrated the ability to facilitate transferability from a data augmentation perspective on the images (Trabucco et al., 2024; You et al., 2024; He et al., 2023). Unlike the traditional hand-crafted data augmentation methods, diffusion models can produce more diverse patterns with high quality Trabucco et al. (2024). With the aid of diffusion guidance (Ho & Salimans, 2022; Dhariwal & Nichol, 2021), these methods can achieve domain customization tailored to specific semantic spaces (You et al., 2024; He et al., 2023). Despite the success of data augmentation through diffusion models on images, the non-Euclidean nature of graph structures poses challenges for data-centric learning on graphs. In addition, the fact that most graphs in the Network Repository are unlabeled exacerbates the challenges, as the absence of labeled data results in substantially lower generation quality for diffusion models (Dhariwal & Nichol, 2021; Bao et al., 2022).

To address the aforementioned challenges, we propose to construct a self-conditioned discrete diffusion model on graph structures. Unlike Gaussian-based diffusion models, discrete diffusion models (Hoogeboom et al., 2021; Austin et al., 2021; Campbell et al., 2022; Vignac et al., 2023) operate with discrete transition kernels between latent variables, as shown in Section 2. The key reason we opt for the discrete diffusion models lies in the sparse nature of graphs, where adding Gaussian noise into the adjacency matrix will result in a dense graph (Haefeli et al., 2022). On the contrary, discrete diffusion models effectively preserve the sparse structure of graphs during the diffusion process, thus maintaining the efficiency of the models on graphs.

To accommodate for unlabeled graphs, we adopt a self-supervised labeling strategy as an auxiliary conditioning procedure (Gao et al., 2022; Hu et al., 2023). By leveraging the self-labeling technique, we are able to upscale the diffusion model to data with more diverse patterns (Gao et al., 2022). The self-labeling technique requires two components: a feature extractor and a self-supervised annotator.

**Feature extractor.** We extract graph-level features by calculating graph properties, including the number of nodes, density, network entropy, average degree, degree variance, and scale-free exponent following Xu et al. (2023). The first two represent the scale of the graph corresponding to nodes and

edges, and the rest indicate the amount of information contained within a graph (Xu et al., 2023). We compute the properties of one graph and concatenate them to get a graph-level representation.

**Self-supervised annotator.** To assign labels to graphs in a self-supervised manner, we employ clustering algorithms on the graph-level representations. The number of clusters is determined jointly by the silhouette score (Rousseeuw, 1987) and the separation of the graphs. The candidates of the number of clusters are chosen to ensure different clusters are well separated. Among the candidates, we select the final number of clusters by maximizing the mean Silhouette Coefficient of all samples.

Next we detail the parameterization of the denoising model $\tilde{p}_\theta(\widetilde{\mathbf{A}}^0 \mid \mathbf{A}^t)$ with the self-assigned graph-level labels $\mathbf{k}$. The denoising model recovers the edges of the original adjacency matrix by predicting the connectivity of the upper triangle, which can be formulated as a link prediction problem (Zhang & Chen, 2018; Kumar et al., 2020). Following the link prediction setup, the denoising model is composed of a graph transformer (GT) (Shi et al., 2020) and an MLP link predictor. Denote the hidden dimension as $d$, we treat the node degrees as node features and utilize a linear mapping $f_d : \mathbb{R} \mapsto \mathbb{R}^d$ to match the dimension. Similarly, we utilize another linear mapping $f_t : \mathbb{R} \mapsto \mathbb{R}^d$ for timestep $t$ and learnable embeddings $f_k : \{0, \ldots, K\} \mapsto \mathbb{R}^d$ for labels $\mathbf{k}$, where $K$ is the number of clusters. The outputs are summed together and then fed into the GT. Mathematically, we have

$$
\begin{aligned}
\mathbf{h}^t &= \mathrm{GT}\left(f_d\left(\mathrm{degree}\left(\mathbf{A}^t\right)\right) + f_t(t) + f_k(\mathbf{k}), \mathbf{A}^t\right), \\
\tilde{p}_\theta(\widetilde{\mathbf{A}}^0_{ij} \mid \mathbf{A}^t; t, \mathbf{k}) &:= \tilde{p}_\theta(\widetilde{\mathbf{A}}^0_{ij} \mid \mathbf{h}^t) = \mathrm{MLP}\left([\mathbf{h}^t_i, \mathbf{h}^t_j]\right).
\end{aligned}
\tag{4}
$$

With the above denoising network, our diffusion model is trained on the collected graphs by optimizing the variational lower bound in (3). After the pre-training process, we perform adaptive downstream enhancement through graph structure augmentation.

## 3.3 Downstream adaptation through data augmentation

The downstream phase of *UniAug* is to augment the graph topology through guided generation. This guidance process serves to provide downstream semantics for the diffusion model, thus bridging the gap between the pre-training distribution and the downstream datasets. Among the techniques for diffusion guidance, gradient-based methods (Dhariwal & Nichol, 2021; Gruver et al., 2024) offer versatile approaches by incorporating external conditions that are not present during training. For the discrete diffusion process, we opt for the gradient-based NOS method (Gruver et al., 2024) due to its flexibility and efficiency. Specifically, we build an *MLP regression head* $g_\theta : \mathbb{R}^d \mapsto \mathbb{R}^r$ that takes the hidden representations $\mathbf{h}^t$ as the input and outputs the guidance objective of dimension $r$. Denote $\tau$ as the temperature, $\gamma$ as the step-size, $\lambda$ as the regularization strength, and $\varepsilon$ drawn from $\mathcal{N}(0, I)$, we sample from $\tilde{p}'(\widetilde{\mathbf{A}}^0 \mid \mathbf{h}^t) \propto \tilde{p}_\theta(\widetilde{\mathbf{A}}^0 \mid \mathbf{h}^t) \exp\left(g_\theta\left(\mathbf{h}^t\right)\right)$ via Langevin dynamics

$$
\mathbf{h}^{t,\prime} \leftarrow \mathbf{h}^{t,\prime} - \gamma\nabla_{\mathbf{h}^{t,\prime}}\left[\lambda\mathrm{KL}\left(\tilde{p}'\left(\widetilde{\mathbf{A}}^0 \mid \mathbf{h}^{t,\prime}\right) \| \tilde{p}'\left(\widetilde{\mathbf{A}}^0 \mid \mathbf{h}^t\right)\right) - g_\theta\left(\mathbf{h}^{t,\prime}\right)\right] + \sqrt{2\gamma\tau}\varepsilon.
\tag{5}
$$

One key question to answer is how to choose the proper guidance objectives. Our goal is to find numerical characteristics that can best describe the structural properties of a graph. This includes supervision signal and self-supervised information on the level of node, edge, and graph.

**Node level.** Node labels provide the supervision signal for node classification tasks. Beyond node labels, node degrees are a fundamental factor in the evolutionary process of a graph (Liu et al., 2011). From the perspective of network analysis, centrality measures indicate the importance of nodes from various viewpoints (Borgatti, 2005). Empirically, we observe that utilizing different node-level heuristics as guidance targets tends to yield similar outcomes. Therefore, we focus on node labels and node degrees.

**Edge level.** Edge-level heuristics can be broadly classified into two categories: local structural heuristics, such as Common Neighbor and Adamic Adar (Adamic & Adar, 2003), and global structural heuristics, such as Katz (Katz, 1953) and SimRank (Jeh & Widom, 2002). Similar to node-level heuristics, empirical observations suggest that different edge-level heuristics tend to yield comparable guidance effects. In this work, we focus on the Common Neighbors (CN) heuristic due to its efficiency. Another edge-level guidance objective is to recover the adjacency matrix from the node representations in a link prediction way, similar to how we parameterize the denoising network. We anticipate that such link prediction objective helps to align the generated graph with the downstream data on the granularity of edges.

Table 1: Comparison between GDA methods, pre-training methods, and *UniAug*. By cross-domain transfer, we emphasize the ability of the method to train on vastly different domains and benefit all of them.

| | GDA methods | | | | Pre-training methods | | | *UniAug* |
|---|---|---|---|---|---|---|---|---|
| | GraphAug | CFLP | Half-Hop | FLAG | AttrMask | D-SLA | GraphMAE | |
| Effective on graph-level task | ✓ | – | – | ✓ | ✓ | ✓ | ✓ | ✓ |
| Effective on edge-level task | – | ✓ | – | ✓ | – | ✓ | – | ✓ |
| Effective on node-level task | – | – | ✓ | ✓ | – | – | ✓ | ✓ |
| In-domain transfer | – | – | – | – | ✓ | ✓ | ✓ | ✓ |
| Cross-domain transfer | – | – | – | – | – | – | – | ✓ |

**Graph level.** Graph labels offer the supervision signal for graph classes or regression targets. In addition, we incorporate graph-level properties (Xu et al., 2023) as quantitative measures to bridge the gap between the pre-training distribution and the downstream dataset. We empirically observe that graph label guidance offers significantly higher performance boosts compared to properties on graph-level tasks. Therefore, we focus on graph labels in our experiments.

We provide our choice of objectives for each task in Appendix B. We note that all the above objectives are natural choices inspired by heuristics and downstream tasks. There exist many other self-supervised objectives to be explored, such as community-level spectral change (Tan et al., 2024) and motif occurrence prediction (Rong et al., 2020). We leave the study of objectives as one future work. With the diffusion guidance, we assemble the augmented graphs with *generated structures* and *original node features*. The augmented graphs are then fed into downstream-specific GNNs.

### 3.4 COMPARISON TO EXISTING METHODS

The data augmentation paradigm of *UniAug* allows us to disentangle the upstream and downstream. We construct a diffusion model as the upstream component to comprehend the structural patterns of graphs across various domains. In addition, we leverage downstream inductive biases with downstream-specific models in a plug-and-play manner. This allows *UniAug* to facilitate cross-domain transfer, offering a unified method that benefits graphs across different domains for various downstream tasks. On the contrary, existing GDA methods are typically designed for specific tasks and hard to transfer to unseen patterns. In the meantime, existing pre-training methods fail to transfer across domains due to heterogeneity in features and structures. This comparison highlights the success of *UniAug* as a data-scaling graph structure augmentor across domains. We summarize the comparison between methods in Table 1.

## 4 EXPERIMENT

In this section, we conduct experiments to validate the effectiveness of *UniAug*. We first pre-train our discrete diffusion model on thousands of graphs collected from diverse domains. For each downstream task, we train an *MLP guidance head* with corresponding objectives on top of the diffusion model. We then perform structure augmentation using *UniAug* and subsequently train a task-specific GNN on augmented data for prediction. Through the experiments, we aim to answer the following research questions:

- RQ1: Can *UniAug* benefit graphs from various domains across different downstream tasks?
- RQ2: What is the scaling behavior of *UniAug* corresponding to data scale and amount of compute?
- RQ3: Which components of *UniAug* are effective in preventing negative transfer?

### 4.1 MAIN RESULTS

To get a comprehensive understanding of *UniAug*, we evaluate it on 25 downstream datasets from 7 domains for graph property prediction, link prediction, and node classification. The statistics of the datasets can be found in Appendix C, and technical details of the experiments are in Appendix D.

**Baselines.** We evaluate our model against three main groups of baselines. (1) Task-specific GNNs: For graph property prediction, we use GIN (Xu et al., 2018); for link prediction, we use GCN (Kipf & Welling, 2017) and NCN (Wang et al., 2024); and for node classification, we use GCN (Kipf & Welling, 2017). (2) Graph pre-training methods: These include AttrMask, CtxtPred, EdgePred, and

Table 2: Mean and standard deviation of accuracy (%) with 10-fold cross-validation on graph classification. The best result is **bold**. The  highlighted  results indicate negative transfer for pre-training methods compared to GIN. The last column is the average rank of each method.

| | DD | Enzymes | Proteins | NCI1 | IMDB-B | IMDB-M | Reddit-B | Reddit-12K | Collab | A.R. |
|---|---|---|---|---|---|---|---|---|---|---|
| GIN | 75.81 ± 6.11 | 66.00 ± 7.52 | 73.32 ± 4.03 | 78.30 ± 3.20 | 71.10 ± 2.90 | 49.07 ± 2.81 | 90.85 ± 1.30 | 48.63 ± 1.62 | 74.54 ± 2.41 | 5.56 |
| AttrMask | 72.93 ± 3.09 | 23.66 ± 6.09 | 73.10 ± 3.90 | 77.67 ± 2.53 | 71.20 ± 2.40 | 48.00 ± 3.14 | 87.50 ± 3.31 | 48.00 ± 1.60 | 75.64 ± 1.52 | 8.00 |
| CtxtPred | 75.14 ± 2.67 | 21.67 ± 3.87 | 72.21 ± 4.60 | 78.99 ± 1.29 | 70.70 ± 1.55 | 48.20 ± 2.23 | 90.35 ± 2.31 | 47.62 ± 2.50 | 75.60 ± 1.49 | 7.67 |
| EdgePred | 75.64 ± 2.77 | 22.00 ± 3.32 | 71.22 ± 3.53 | 77.82 ± 2.95 | 70.20 ± 2.23 | 47.80 ± 2.42 | 90.80 ± 1.69 | 48.35 ± 1.44 | 74.64 ± 2.24 | 8.56 |
| InfoMax | 75.23 ± 3.43 | 22.50 ± 6.76 | 71.30 ± 5.18 | 76.94 ± 1.48 | 71.60 ± 2.06 | 46.70 ± 2.46 | 89.15 ± 2.84 | 48.98 ± 1.83 | 75.44 ± 1.12 | 8.00 |
| JOAO | 75.98 ± 2.86 | 22.17 ± 3.67 | 71.57 ± 5.31 | 76.87 ± 2.27 | 71.02 ± 1.81 | 48.85 ± 2.06 | 90.17 ± 2.13 | 49.01 ± 1.90 | 74.77 ± 1.71 | 7.11 |
| D-SLA | 74.66 ± 3.30 | 22.67 ± 4.21 | 71.97 ± 4.17 | 77.95 ± 2.11 | 71.92 ± 2.75 | 47.28 ± 1.88 | 89.77 ± 1.87 | 48.50 ± 1.33 | 75.99 ± 2.08 | 7.00 |
| GraphMAE | 76.07 ± 3.25 | 23.00 ± 3.64 | 70.45 ± 4.19 | 79.08 ± 2.72 | 71.50 ± 2.01 | 47.93 ± 3.03 | 86.10 ± 3.63 | 47.67 ± 1.16 | 74.84 ± 1.36 | 7.67 |
| S-Mixup | 73.12 ± 3.27 | 66.85 ± 7.04 | 74.61 ± 5.08 | 78.91 ± 1.61 | 69.61 ± 4.43 | 48.33 ± 5.36 | 88.65 ± 3.12 | 48.30 ± 2.50 | 75.89 ± 3.26 | 6.67 |
| GraphAug | 75.21 ± 2.63 | 68.14 ± 7.92 | 74.21 ± 3.70 | 79.53 ± 3.21 | **74.00 ± 3.41** | 48.11 ± 1.85 | 90.50 ± 3.17 | 49.00 ± 1.99 | 76.02 ± 2.67 | 3.67 |
| FLAG | 76.87 ± 7.21 | 68.35 ± 7.45 | 74.31 ± 4.21 | 79.03 ± 3.75 | 68.83 ± 4.67 | 47.21 ± 3.45 | 89.11 ± 2.40 | 47.48 ± 3.01 | 75.32 ± 3.13 | 7.00 |
| *UniAug* | **78.13 ± 2.61** | **71.50 ± 5.85** | **75.47 ± 2.50** | **80.54 ± 1.77** | 73.50 ± 2.48 | **50.13 ± 2.05** | **92.28 ± 1.59** | **49.48 ± 0.71** | **77.00 ± 2.02** | 1.11 |

InfoMax (Hu et al., 2019), JOAO (You et al., 2021), D-SLA (Kim et al., 2022), and GraphMAE (Hou et al., 2022). For each of these methods, we pre-train it on the same pre-training set as *UniAug*. While most of the pre-training graphs lack node features, we calculate the node degrees as the input. Each method consists of three pre-trained variants with different backbone GNNs, including GIN, GCN, and GAT. We note that all these methods require the downstream graphs to have the same node feature space as the pre-training data. Therefore, in the fine-tuning stage, we replace the node features of the downstream datasets with node degrees, evaluate all three variants, and report the highest performance for each method in each task. We are aware that simply using the node degrees could lead to a decline in performance for the baseline methods. Thus, we include more results with *semi-supervised* and *self-supervised* settings in Appendix D. (3) Graph data augmentation (GDA) methods: For graph property prediction, we include S-Mixup (Ling et al., 2023), GraphAug (Luo et al., 2022), FLAG (Kong et al., 2022), GREA (Liu et al., 2022a), and DCT (Liu et al., 2024a); for link prediction, we include CFLP (Zhao et al., 2022); and for node classification on heterophilic graphs, we include Half-Hop (Azabou et al., 2023). The GDA methods are implemented based on chosen task-specific GNNs.

**Graph property prediction.** We employ graph label guidance for *UniAug* throughout the graph-level tasks by training a 2-layer MLP as the guidance head on the graph labels in the training set. In the augmentation stage, we generate multiple graphs per training sample, and the generated graphs are then fed into the baseline GIN. We present the results of molecule regression in Table 3 and graph classification in Table 2. Three key observations emerge from the analysis: (1) Existing pre-training methods show negative transfer compared to GIN. Some special cases are the Enzymes and molecule regression datasets, where all pre-training methods fail to yield satisfactory results. In these datasets, the features are one of the driving components

Table 3: Mean and standard deviation of MAE ↓ across 10 runs on molecule regression. The last column is the average rank of each method. Among the methods, all pre-training methods discard atom and bond features due to dimension mismatch and we include the best-performing method JOAO into comparison; GIN and *UniAug* remove the bond features; others incorporate both.

| | ogbg-Lipo | ogbg-ESOL | ogbg-FreeSolv | A.R. |
|---|---|---|---|---|
| GINE* | 0.545 ± 0.019 | 0.766 ± 0.016 | 1.639 ± 0.146 | 5.00 |
| GIN | 0.543 ± 0.021 | 0.729 ± 0.018 | 1.613 ± 0.155 | 3.67 |
| JOAO | 0.859 ± 0.007 | 1.458 ± 0.040 | 3.292 ± 0.117 | 7.00 |
| FLAG* | 0.528 ± 0.012 | 0.755 ± 0.039 | 1.565 ± 0.098 | 3.00 |
| GREA* | 0.586 ± 0.036 | 0.805 ± 0.135 | 1.829 ± 0.368 | 6.00 |
| DCT* | 0.516 ± 0.071 | 0.717 ± 0.020 | 1.339 ± 0.075 | 1.33 |
| *UniAug* | 0.528 ± 0.006 | 0.677 ± 0.026 | 1.448 ± 0.049 | 1.67 |

*Results are taken from DCT (Liu et al., 2024a).

for graph property prediction, while the pre-training methods fail to encode such information due to incompatibility with the feature dimension. This reveals one critical drawback of the pre-training methods: their inability to handle feature heterogeneity. (2) GDA methods yield inconsistent results across different datasets. While these methods enhance performance in some datasets, they cause performance declines in others. This variability is directly reflected in the average rank, where some of them even fall behind the GIN. (3) Unlike the pre-training methods and GDA methods, *UniAug* shows consistent performance improvements against GIN with a large margin. In the molecule regression tasks, *UniAug* effectively compensates for the absence of bond features and achieves performance comparable to DCT, which is a data augmentation method pre-trained on in-domain molecule graphs. Note that we replace the original node features with node degrees when pre-training the baselines on our graph collection due to missing features and mismatched semantics. We understand that removing

Table 4: Mean and standard deviation across 10 runs on link prediction. Results are scaled ×100. The last two methods are based on NCN, while the rest are GCN-based. The best result is **bold** for two backbones, respectively. The highlighted results indicate negative transfer for pre-training methods compared to GCN. The last column is the average rank of each GCN-based method.

| | Cora MRR | Citeseer MRR | Pubmed MRR | Power Hits@10 | Yeast Hits@10 | Erdos Hits@10 | Flickr Hits@10 | A.R. |
|---|---|---|---|---|---|---|---|---|
| GCN | 30.26 ± 4.80 | 50.57 ± 7.91 | 16.38 ± 1.30 | 30.61 ± 4.07 | 24.71 ± 4.92 | 35.71 ± 2.65 | 8.10 ± 2.58 | 4.14 |
| AttrMask | 13.43 ± 1.93 | 20.23 ± 1.29 | 16.39 ± 3.62 | 29.92 ± 2.61 | 25.10 ± 4.77 | 30.85 ± 3.13 | 8.77 ± 1.65 | 6.43 |
| CtxtPred | 15.68 ± 2.91 | 22.31 ± 1.31 | 13.10 ± 3.70 | 29.30 ± 3.55 | 22.96 ± 4.28 | 34.82 ± 2.55 | 3.61 ± 1.01 | 7.86 |
| EdgePred | 15.31 ± 3.54 | 22.91 ± 1.87 | 17.85 ± 4.45 | 29.54 ± 3.78 | 25.78 ± 4.51 | 34.65 ± 3.84 | 6.86 ± 3.24 | 5.43 |
| InfoMax | 16.35 ± 2.57 | 22.90 ± 1.30 | 15.91 ± 2.71 | 29.29 ± 4.72 | 26.33 ± 4.12 | 35.82 ± 4.12 | 3.23 ± 0.38 | 6.00 |
| JOAO | 17.21 ± 3.66 | 23.10 ± 1.41 | 15.33 ± 3.70 | 28.98 ± 4.01 | 26.47 ± 4.65 | 33.77 ± 3.05 | 6.01 ± 1.57 | 6.00 |
| D-SLA | 15.55 ± 3.12 | 23.05 ± 1.54 | 16.10 ± 3.96 | 29.37 ± 2.88 | 26.15 ± 3.32 | 36.02 ± 4.58 | 6.70 ± 2.03 | 5.29 |
| GraphMAE | 15.94 ± 1.73 | 20.35 ± 1.52 | 13.80 ± 1.36 | 27.69 ± 1.99 | 26.51 ± 2.92 | 35.63 ± 3.61 | 8.41 ± 2.44 | 6.14 |
| CFLP | 33.62 ± 6.44 | **55.20 ± 4.16** | 17.01 ± 2.75 | 16.02 ± 8.31 | 24.23 ± 5.23 | 28.74 ± 2.38 | OOM | 6.43 |
| *UniAug*-GCN | **35.36 ± 7.88** | 54.66 ± 4.55 | **17.28 ± 1.89** | **34.36 ± 1.68** | 27.52 ± 4.80 | **39.67 ± 4.51** | **9.46 ± 1.18** | 1.29 |
| NCN | 31.72 ± 4.48 | 58.03 ± 3.45 | 38.26 ± 2.56 | 27.36 ± 5.00 | 39.85 ± 5.07 | 36.81 ± 3.29 | 8.33 ± 0.92 | – |
| *UniAug*-NCN | **35.92 ± 7.85** | **61.69 ± 3.21** | **40.30 ± 2.53** | **30.20 ± 1.46** | **42.11 ± 5.74** | **39.26 ± 2.84** | **8.85 ± 0.90** | – |

Table 5: Mean and standard deviation of accuracy (%) across 10 splits on node classification of heterophilic graphs. The best result is **bold**. The highlighted results indicate negative transfer for pre-training methods compared to GCN. The last column is the average rank of each method.

| | Cornell | Wisconsin | Texas | Actor | Chameleon* | Squirrel* | A.R. |
|---|---|---|---|---|---|---|---|
| GCN | 59.41 ± 6.03 | 51.68 ± 4.34 | 63.78 ± 4.80 | 30.58 ± 1.29 | 40.94 ± 3.91 | 39.11 ± 1.74 | 3.83 |
| AttrMask | 44.86 ± 5.43 | 53.73 ± 4.31 | 60.54 ± 5.82 | 25.31 ± 1.03 | 35.81 ± 2.88 | 30.63 ± 1.68 | 5.83 |
| CtxtPred | 40.81 ± 7.78 | 36.67 ± 17.23 | 58.92 ± 4.32 | 23.97 ± 2.63 | 24.36 ± 4.13 | 26.26 ± 7.50 | 9.50 |
| EdgePred | 42.70 ± 5.51 | 48.04 ± 6.63 | 59.37 ± 5.11 | 22.99 ± 6.22 | 21.02 ± 5.06 | 27.94 ± 8.41 | 8.83 |
| InfoMax | 39.19 ± 12.75 | 39.80 ± 16.38 | 58.87 ± 4.06 | 23.30 ± 4.37 | 22.59 ± 4.91 | 27.52 ± 9.09 | 10.17 |
| JOAO | 40.13 ± 8.60 | 44.70 ± 7.45 | 57.06 ± 3.43 | 24.17 ± 5.02 | 25.81 ± 3.79 | 31.72 ± 7.03 | 8.33 |
| D-SLA | 41.05 ± 6.88 | 42.13 ± 9.58 | 59.93 ± 4.29 | 23.74 ± 4.06 | 26.49 ± 4.27 | 28.50 ± 6.90 | 8.00 |
| GraphMAE | 47.05 ± 4.37 | 57.06 ± 4.59 | 63.70 ± 5.51 | 24.69 ± 0.68 | 37.18 ± 3.08 | 31.94 ± 1.65 | 5.00 |
| Half-Hop | 62.46 ± 7.58 | 76.47 ± 2.61 | 72.35 ± 4.27 | 33.95 ± 0.68 | 38.59 ± 2.89 | 37.34 ± 2.18 | 3.00 |
| *UniAug* | 68.11 ± 6.72 | 69.02 ± 4.96 | 73.51 ± 5.06 | 33.11 ± 1.57 | **43.84 ± 3.39** | **41.90 ± 1.90** | 2.00 |
| *UniAug* + Half-Hop | **72.43 ± 5.81** | **79.61 ± 5.56** | **77.03 ± 4.27** | **34.97 ± 0.55** | 41.94 ± 2.77 | 38.79 ± 2.61 | 1.50 |

*Chameleon and Squirrel are filtered to remove duplicated nodes (Platonov et al., 2023).

the node features may result in a performance drop for the baseline methods. Therefore, we adapt the *semi-supervised* (You et al., 2020) and *self-supervised* (Sun et al., 2020) setting for the baselines for a comprehensive benchmark in Appendix D.1 Table 13, where we observe that UniAug presents consistently satisfactory performance according to the average rank, matching or outperforming the best baseline. These findings affirm that the pre-training and structure augmentation paradigm of *UniAug* effectively benefits the downstream datasets at the graph level.

**Link prediction.** We choose three guidance objectives for *UniAug*, including node degree, CN, and link prediction objective, as described in Section 3.3. For each objective, we train an MLP to provide guidance information. We then augment the graph structure by generating a synthetic graph and preserving the original training edges, ensuring that the augmented graph does not remove any existing edges. The augmented graph is then fed into a GCN for link prediction. We summarize the results in Table 4, which show similar patterns to those observed in graph property prediction: (1) Existing pre-training methods provide negative transfer, especially on datasets with node features. (2) GDA method CFLP leads to performance drops on the datasets without features and also suffers from high computation complexity during preprocessing. (3) *UniAug* enhances performance across all tested datasets. In addition, we employ *UniAug* to NCN (Wang et al., 2024), one of the state-of-the-art methods for link prediction. The results demonstrate consistent performance boosts from *UniAug* when we apply NCN as the backbone. The structure augmentation paradigm of *UniAug* allows plug-and-play applications to any downstream-specific models, showcasing its adaptability and effectiveness. In addition, we study the effects of three guidance objectives. More details can be found in Appendix D.2.

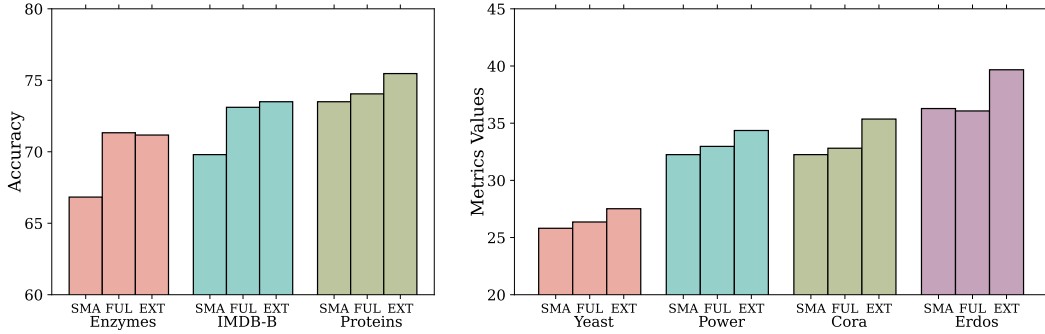

Figure 3: Effects of pre-training data scale on graph classification (left) and link prediction (right). The groups SMA, FUL, and EXT represent SMALL, FULL, and EXTRA data collection.

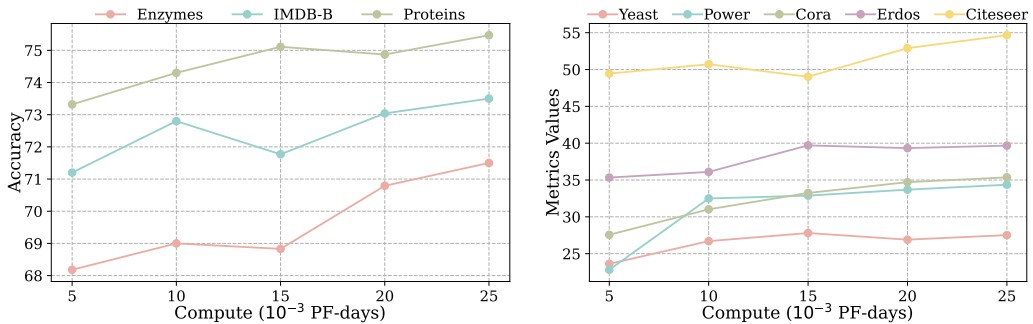

Figure 4: Effects of pre-training amount of compute on graph classification (left) and link prediction (right), where one PF-days $= 10^{15} \times 24 \times 3600 = 8.64 \times 10^{19}$ floating point operations.

**Node classification.** To demonstrate the effectiveness of *UniAug* in node-level tasks, we transform the node classification into subgraph classification. Specifically, we extract the aggregation tree of each node, i.e., 2-hop subgraph for a 2-layer GCN, and label the subgraph with the center node. We then adopt a strategy similar to graph classification and train a 2-layer classifier as a guidance head. Inspired by the success of structure augmentation on heterophilic graphs (Bi et al., 2022; Azabou et al., 2023), we evaluate *UniAug* on 6 heterophilic datasets. We observe phenomena similar to those seen in graph- and link-level tasks in Table 5. One thing to mention is the combination of *UniAug* and Half-Hop. Half-Hop offers performance improvements in four out of six datasets via data augmentation, and combining it with *UniAug* yields even higher results. This highlights the flexibility of *UniAug* and opens up possibilities for further exploration of its use cases. Given the impressive results of *UniAug* on heterophilic graphs, we anticipate it will also help to balance the performance disparities among nodes with different homophily ratios on homophilic graphs (Mao et al., 2024a). We split the nodes into five groups according to their homophily ratios and calculate the standard deviation (SD) across groups. As shown in Table 6, *UniAug* matches the performance of vanilla GCN and also reduces the performance discrepancies corresponding to SD.

Table 6: Results of node classification on homophily graphs. Results are scaled $\times 100$.

|  |  | Cora | Citeseer | Pubmed |
|---|---|---|---|---|
| ACC ↑ | GCN | 81.75 ± 0.73 | 70.71 ± 0.76 | 79.53 ± 0.25 |
|  | *UniAug* | 81.78 ± 0.60 | 71.17 ± 0.58 | 79.54 ± 0.35 |
| SD ↓ | GCN | 24.51 ± 1.06 | 22.57 ± 0.80 | 27.02 ± 0.56 |
|  | *UniAug* | **23.45 ± 0.90** | **19.90 ± 0.81** | **26.50 ± 0.55** |

### 4.2 SCALING BEHAVIOR OF *UniAug*

In light of the neural scaling law (Kaplan et al., 2020; Hoffmann et al., 2022; Abnar et al., 2022; Zhai et al., 2022; Liu et al., 2024b), we expect *UniAug* to benefit from an increased coverage of data and more compute budget. In this subsection, we investigate the scaling behavior of *UniAug* in terms of data scale and amount of compute for pre-training.

**Data coverage** During the data collection process, we prepare three versions of the training data with increasing magnitude and growing coverage on the graph distribution. We first sample 10 graphs

per category from the Network Repository (Rossi & Ahmed, 2015) to build a SMALL collection. Next, we gather all the graphs from the Network Repository and filter out large-scale graphs and outliers for a FULL collection. In addition, we add a 1000 graphs subset of the GitHub Star dataset from TUDataset (Morris et al., 2020) to enlarge the coverage of diverse patterns and form an EXTRA collection. We pre-train three versions of *UniAug* respectively on the three collections and evaluate them on graph classification and link prediction. As shown in Fig. 3, we observe a clear trend of increase in performance as we enlarge the coverage of pre-training data. This paves the way to scale up *UniAug* to even more pre-training graphs with an expanding distribution of graphs.

**Amount of compute** We sought to understand how effectively our diffusion model can learn data patterns as we continue to train it. To this end, we checkpointed *UniAug* every 2,000 epochs ($5 \times 10^{-3}$ PF-days) while training on the EXTRA collection, and then applied it to graph classification and link prediction tasks. The results are illustrated in Fig. 4. We observe that downstream performance generally improves with prolonged training, while the trend slows down for some datasets when we reach 8,000 epochs. We take the checkpoint at the 10,000th epoch for evaluations. Given the scaling behavior observed, we anticipate *UniAug* to become even more effective with additional resources.

## 4.3 PREVENTING NEGATIVE TRANSFER

In the previous parts of the experiments, we showcase the positive transfer of *UniAug* across different tasks. We now investigate which aspects of the design prevent negative transfer. *UniAug* consists of two main components: a pre-trained diffusion model and the structure augmentation through guided generation. In the pre-training process, we inject self-supervised graph labels into the diffusion model and we wonder about the performance of its unconditioned counterpart. Regarding the augmentation process, we examine the impact of diffusion guidance by exploring outcomes when the guidance is either removed or applied using another dataset from a different domain (cross-guide). We summarize the results in Table 7 for graph classification and link prediction. All modifications investigated lead to performance declines in both tasks. We observe that removing guidance results in significant negative transfers for graph classification, while the effects of self-conditioning are more pronounced for link prediction. We conclude that both the self-conditioning strategy and diffusion guidance are crucial in preventing negative transfer, underscoring their importance in the design of *UniAug*.

Table 7: Demonstration of negative transfer on graph classification (up) and link prediction (down).

|  | Enzymes | Proteins | IMDB-B | IMDB-M |
|---|---|---|---|---|
| GIN | 66.00 ± 7.52 | 73.32 ± 4.03 | 71.10 ± 2.90 | 49.07 ± 2.81 |
| *UniAug* | 71.50 ± 5.85 | 75.47 ± 2.50 | 73.50 ± 2.48 | 50.13 ± 2.05 |
| w/o self-cond | 71.11 ± 7.50 | 73.31 ± 4.63 | 71.50 ± 2.27 | 49.00 ± 2.74 |
| w/o guidance | 62.17 ± 3.93 | 71.15 ± 4.56 | 53.80 ± 3.29 | 35.33 ± 3.17 |
| w/ cross-guide | 51.50 ± 7.64 | 72.46 ± 4.35 | 71.10 ± 2.38 | 49.20 ± 2.59 |

|  | Cora MRR | Citeseer MRR | Power Hits@10 | Yeast Hits@10 | Erdos Hits@10 |
|---|---|---|---|---|---|
| GCN | 30.26 ± 4.80 | 50.57 ± 7.91 | 30.61 ± 4.07 | 24.71 ± 4.92 | 35.71 ± 2.65 |
| *UniAug* | 35.36 ± 7.88 | 54.66 ± 4.55 | 34.36 ± 1.68 | 27.52 ± 4.80 | 39.67 ± 4.51 |
| w/o self-cond | 27.97 ± 16.11 | 37.65 ± 6.00 | 28.95 ± 7.73 | 23.54 ± 8.28 | 34.33 ± 6.18 |
| w/o guidance | 29.60 ± 6.06 | 51.41 ± 7.10 | 25.57 ± 6.04 | 25.26 ± 6.06 | 37.11 ± 4.16 |
| w/ cross-guide | 32.37 ± 4.20 | 50.59 ± 5.67 | 32.99 ± 2.54 | 26.76 ± 3.88 | 36.30 ± 3.67 |

## 5 CONCLUSION AND DISCUSSION

In this work, we propose a graph structure augmentation pipeline *UniAug* to leverage the increasing scale of graph data. We collect thousands of graphs from various domains and pre-train a self-conditioned discrete diffusion model on them. In the downstream stage, we augment the graphs by preserving the original node features and generating synthetic structures. We apply *UniAug* to node-, link-, and graph-level tasks and achieve consistent performance gain. We have successfully developed a showcase that benefits from cross-domain graph data scaling using diffusion models.

One limitation of the current analysis is the absence of an investigation into the effects of model parameters due to limited resources. Given the scaling behavior of *UniAug* in terms of data scale and amount of compute, we anticipate that a large-scale model will provide significant performance improvements. One future direction is to investigate the adaptation of fast sampling methods to the discrete diffusion models on graphs. This will lead to lower time complexity and enable broader application scenarios.

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

# A    DERIVATION OF DIFFUSION PROCESS

In the following, we will formulate the existing discrete diffusion models into binary diffusion on the adjacency matrix. We denote the adjacency matrix of a graph as $\mathbf{A}^0 \in \{0,1\}^{n \times n}$ with $n$ nodes. Following D3PM (Austin et al., 2021), we corrupt the adjacency matrix into a sequence of latent variables $\mathbf{A}^{1:T} = \mathbf{A}^1, \mathbf{A}^2, \ldots, \mathbf{A}^T$ by independently injecting noise into each element with a Markov process

$$q\left(\mathbf{A}^t \mid \mathbf{A}^{t-1}\right) = \prod_{i,j:i<j} \mathrm{Cat}\left(\mathbf{A}_{ij}^t; \mathbf{p} = \mathbf{A}_{ij}^{t-1}\mathbf{Q}^t\right), \tag{6}$$

where $\mathbf{Q}^t \in [0,1]^{2 \times 2}$ is the transition probability of timestep $t$. The above Markov process is called *forward process*. Existing works provide different designs for the transition matrix $\mathbf{Q}^t$, including

$$\begin{aligned}
\text{Uniform (Chen et al., 2023a)} &: \begin{pmatrix} 1 - \beta^t & \beta^t \\ \beta^t & 1 - \beta^t \end{pmatrix}; \\
\text{Absorbing (Chen et al., 2023b)} &: \begin{pmatrix} 1 & 0 \\ \beta^t & 1 - \beta^t \end{pmatrix}; \\
\text{Predefined (Vignac et al., 2023)} &: \begin{pmatrix} 1 - \beta^t \cdot \pi & \beta^t \cdot p \\ (1-\pi)\beta^t & 1 - (1-\pi)\beta^t \end{pmatrix},
\end{aligned} \tag{7}$$

where $\pi$ is the converging non-zero probability and $\beta^t$ is the noise scale. All three transition matrices can be written as binary diffusion with Bernoulli distribution

$$\begin{aligned}
q\left(\mathbf{A}^t \mid \mathbf{A}^{t-1}\right) &= \mathrm{Bernoulli}\left(\mathbf{A}^t; \alpha^t \mathbf{A}^{t-1} + \left(1 - \alpha^t\right)\pi\right), \\
q\left(\mathbf{A}^t \mid \mathbf{A}^0\right) &= \mathrm{Bernoulli}\left(\mathbf{A}^t; \bar{\alpha}^t \mathbf{A}^0 + \left(1 - \bar{\alpha}^t\right)\pi\right), \\
q\left(\mathbf{A}^{t-1} \mid \mathbf{A}^t, \mathbf{A}^0\right) &= \frac{q\left(\mathbf{A}^t \mid \mathbf{A}^{t-1}\right)q\left(\mathbf{A}^{t-1} \mid \mathbf{A}^0\right)}{q\left(\mathbf{A}^t \mid \mathbf{A}^0\right)},
\end{aligned} \tag{8}$$

where $\alpha^t = 1 - \beta^t$ and $\bar{\alpha}^t = \prod_{i=1}^{t} \alpha^i$. The prior $\mathbf{A}^T$ is determined by $\pi$ with $p\left(\mathbf{A}_{ij}^T\right) = \mathrm{Bernoulli}(\pi)$, i.e., the existence of each edge follows a Bernoulli distribution with probability $\pi$. The main difference of the *forward process* among the existing works is the choice of $\pi$, where $\pi = 0$ for EDGE (Chen et al., 2023b), $\pi = 0.5$ for D4Explainer (Chen et al., 2023a), and a pre-computed average density $\pi$ for DiGress (Vignac et al., 2023).

In our early experiments, we observe that the absorbing kernel $\pi = 0$ surpasses the other two in terms of efficiency and effectiveness for graph generation. The *forward process* with non-zero $\pi$ will add non-existing edges, which brings in additional computations. When sampling from prior, non-zero $\pi$ will introduce additional uncertainty because we will first sample every edge from $\mathrm{Bernoulli}(\pi)$. Therefore, we choose the absorbing prior $\pi = 0$ in this work and leave the exploration of other transition kernels as a future work.

We note that in our implementation, we choose the number of timesteps $T$ as 128 according to our early experiments and some existing works (Wang et al., 2023; Chen et al., 2023b). We leave the study of the effects of diffusion timesteps on downstream tasks as a future work.

# B    GUIDANCE OBJECTIVE FOR DOWNSTREAM TASKS

We mention various guidance objectives in Section 3.3 with different granularity. Here, we specify the objectives we use for each downstream task. Our empirical results suggest that supervision signals will lead to better performance. Thus, we use node labels for node classification and graph labels for graph property prediction in Section 4. Regarding link prediction, we anticipate that both node-level and edge-level objectives may help the downstream adaptation. Therefore, we choose three objectives including node degree, CN heuristic, and link prediction objective.

# C    DATASETS

The license of the datasets use in this work is in Table 8.

Table 8: List of datasets and corresponding License

| Dataset | License |
|---|---|
| Network Repository | CC BY-SA |
| Github Star | CC BY 4.0 |
| Cora | NLM license |
| Citeseer | NLM license |
| Pubmed | NLM license |
| WebKB | MIT license |
| Wikipedia Network | MIT license |
| Actor | MIT license |
| Power | BSD License |
| Yeast | BSD License |
| Erdos | BSD License |
| Amazon Photo | MIT license |
| Flickr | MIT license |
| DD | CC BY 4.0 |
| Enzymes | CC BY 4.0 |
| Proteins | CC BY 4.0 |
| NCI1 | CC BY 4.0 |
| IMDB | CC BY 4.0 |
| Reddit | CC BY 4.0 |

**Graph property prediction datasets** include DD and Proteins (Dobson & Doig, 2003), Enzymes (Schomburg et al., 2004), NCI1 (Wale et al., 2008), IMDB-Binary, IMDB-Multi, Reddit-Binary, and Reddit-Multi-12K (Yanardag & Vishwanathan, 2015), ogbg-Lipo, ogbg-ESOL and ogbg-FreeSolv (Hu et al., 2020). The statistics are summarized in 9.

Table 9: Statistics of graph property prediction datasets.

| Domain | Dataset | Task type | # Graphs | # Tasks | # Nodes | # Edges |
|---|---|---|---|---|---|---|
| Biology | DD | Classification | 1,178 | 2 | 284 | 716 |
| | Enzymes | Classification | 600 | 6 | 33 | 64 |
| | Proteins | Classification | 1,113 | 2 | 40 | 73 |
| Academic | Collab | Classification | 5,000 | 3 | 74 | 2458 |
| Social | IMDB-B | Classification | 1,000 | 2 | 20 | 97 |
| | IMDB-M | Classification | 1,500 | 3 | 13 | 66 |
| | Reddit-5k | Classification | 4,999 | 5 | 509 | 595 |
| | Reddit-12k | Classification | 11,929 | 11 | 391 | 1305 |
| Chemical | NCI | Classification | 4,110 | 2 | 30 | 32 |
| | ogbg-Lipo | Regression | 4200 | 1 | 27 | 59 |
| | ogbg-ESOL | Regression | 1128 | 1 | 13 | 27 |
| | ogbg-FreeSolv | Regression | 642 | 1 | 9 | 17 |

**Link prediction datasets** include Cora, Citeseer, and Pubmed (Sen et al., 2008), Power (Watts & Strogatz, 1998), Yeast (Bu et al., 2003), Erdos (Batagelj & Mrvar, 2006), Amazon Photo (Shchur et al., 2018), and Flickr (Leskovec & Krevl, 2014). The statistics are summarized in 10.

Table 10: Statistics of link prediction datasets.

| | Cora | Citeseer | Pubmed | Power | YST | ERD | Flickr |
|---|---|---|---|---|---|---|---|
| Domain | | Citation | | Transport | Biology | Academic | Social |
| #Nodes | 2,708 | 3,327 | 18,717 | 4,941 | 2,284 | 6,927 | 334,863 |
| #Edges | 5,278 | 4,676 | 44,327 | 6,594 | 6,646 | 11,850 | 899,756 |
| Mean Degree | 3.9 | 2.81 | 4.74 | 2.67 | 5.82 | 3.42 | 5.69 |

**Node classification datasets** include Cora, Citeseer, and Pubmed (Sen et al., 2008), WebKB (Texas, Cornell, and Wisconsin) (Pei et al., 2020), Wikipedia Network (Chameleon and Squirrel) (Pei et al., 2020), and Actor (Tang et al., 2009). The first three are homophilic graphs, and the others are heterophilic. The statistics are summarized in 11.

Table 11: Statistics of node classification datasets.

|  | Cora | Citeseer | Pubmed | Cornell | Wisconsin | Texas | Chameleon* | Squirrel* | Actor |
|---|---|---|---|---|---|---|---|---|---|
| Domain | | Citation | | | | Web | | | Social |
| #Nodes | 2,708 | 3,327 | 19,717 | 183 | 251 | 183 | 890 | 2,223 | 7,600 |
| #Edges | 5,278 | 4,676 | 44,324 | 295 | 499 | 309 | 8,854 | 46,998 | 33,544 |
| #Classes | 7 | 6 | 3 | 5 | 5 | 5 | 5 | 5 | 5 |

*Chameleon and Squirrel are filtered to remove duplicated nodes (Platonov et al., 2023).

## D  EXPERIMENT

In this section, we introduce the implementation details and additional results for the experiments. Throughout all the experiments, we train all the methods with Adam optimizer on an A100 GPU. We train the guidance head of *UniAug* with cross-entropy loss for class labels and mean squared error loss for all other objectives. For multi-class objectives, we apply the label smoothing (Szegedy et al., 2016) technique following NOS (Gruver et al., 2024). Denote $y$ as the one-hot label and $C$ as the number of classes, we have

$$\overline{\mathbf{y}}_t = \bar{\alpha}_t * \mathbf{y} + (1 - \bar{\alpha}_t) / C * \mathbf{1}. \tag{9}$$

### D.1  GRAPH PROPERTY PREDICTION

For graph classification, we follow (Errica et al., 2020) for the setting with 10-fold cross-validation. We utilize a 5-layer GIN with latent dimensions of 64 throughout the datasets. For molecule regression, we implement a 5-layer GIN with a virtual node, and the latent dimensions are 300. We have mainly four hyperparameters for *UniAug*: step-size $\gamma$ and regularization strength $\lambda$ in (5), number of repeats per training graph, and whether augment validation and test graphs with the trained guidance head. For each training graph, we repeatedly generate structures and plug in the original node features for multi-repeat augmentation. We perform the update in (5) for 5 times per each sampling step. The hyperparameters are tuned from the choices in Table 12.

Table 12: Hyperparameter choices for graph property prediction.

| | |
|---|---|
| $\lambda$ | 0.01 |
| $\gamma$ | [0.1, 0.5, 1.0] |
| # repeats | [1, 5, 10, 32, 64] |
| Aug val and test | [True, False] |

In Section 4.1, we aim to benchmark the capability of cross-domain pre-training of different methods on the same set of pre-training graphs. While the pre-training graphs contain vastly different features, we have to align the feature space to allow pre-training for the baseline methods. There are two ways to tackle the feature heterogeneity issues in the existing literature. One line of them utilizes LLMs to align text-space graphs (Chen et al., 2024), which is not applicable to broader classes of graphs. Other works, like GCOPE (Zhao et al., 2024), perform dimension reduction to align the feature dimension of different graphs. We emphasize that dimension reduction methods fail to deal with extreme cases like missing features. This phenomenon is pretty common in real life, as a large proportion of the graphs in the Network Repository do not have corresponding features. Therefore, we simply use the node degrees as the features in Section 4.1.

We understand that removing the node features may result in a performance drop for the baseline methods. Note that most of the baselines follow the pre-training paradigm of (Hu et al., 2019) with domain-specific model designs for chemistry and biology datasets, and thus cannot be directly applied to the chosen graph classification datasets. Therefore, we adapt the **semi-supervised** (You et al., 2020) and **self-supervised** (Sun et al., 2020) setting for the baselines for a comprehensive benchmark. The semi-supervised setting involves pre-training with all data of that specific dataset and finetuning the training set of each split. Meanwhile, baselines of the self-supervised setting pre-train on the whole dataset and then classify the learned graph embeddings with a downstream SVM classifier. The results are summarized in Table 13, where the best and second-best results are highlighted in **bold** and *italic*, respectively. We observe that UniAug presents consistently satisfactory performance according to the average rank, matching or outperforming the best baseline.

Table 13: Mean and standard deviation of accuracy (%) with 10-fold cross-validation on graph classification. The best and second-best results are highlighted in **bold** and *italic*. The last column is the average rank.

| | | DD | Proteins | NCI1 | IMDB-B | IMDB-M | Reddit-B | Collab | A.R. |
|---|---|---|---|---|---|---|---|---|---|
| Semi-supervised | CtxtPred | 74.66±0.51 | 70.23±0.63 | 73.00±0.30 | – | – | 88.66±0.95 | 73.69±0.37 | 6.80 |
| | InfoMax | 75.78±0.34 | 72.27±0.40 | 74.86±0.26 | – | – | 88.66±0.95 | 73.76±0.29 | 5.60 |
| | GraphCL | *76.17±1.37* | 74.17±0.34 | 74.63±0.25 | – | – | 89.11±0.19 | 74.23±0.21 | 4.60 |
| | JOAO | 75.81±0.73 | 73.31±0.48 | 74.86±0.39 | – | – | 88.79±0.65 | 75.53±0.18 | 4.60 |
| Self-supervised | InfoGraph | – | 74.44±0.31 | 76.20±1.06 | 73.03±0.87 | 49.69±0.53 | 82.50±1.42 | 70.65±1.13 | 5.17 |
| | GraphCL | – | 74.39±0.45 | 77.87±0.41 | 71.14±0.44 | 48.58±0.67 | *89.53±0.84* | 71.36±1.15 | 4.50 |
| | JOAO | – | 74.55±0.41 | 78.07±0.47 | 70.21±3.08 | 49.20±0.77 | 85.29±1.35 | 69.50±0.36 | 5.17 |
| | GraphMAE | – | *75.30±0.39* | *80.40±0.30* | **75.52±0.66** | **51.63±0.52** | 88.01±0.19 | **80.32±0.46** | 2.17 |
| | UniAug | **78.13±2.61** | **75.47±2.50** | **80.54±1.77** | *73.50±2.48* | *50.13±2.05* | **92.28±1.59** | *77.00±2.02* | 1.43 |

## D.2 LINK PREDICTION

For link prediction, we follow the model designs and evaluation protocols of (Li et al., 2024). For results based on GCN and NCN, we use a GCN encoder to produce node embeddings and perform link prediction with a prediction head. The prediction head of GCN is a 3-layer MLP. The number of layers and the latent dimension of the GCN encoder are taken from (Li et al., 2024). We have mainly three hyperparameters for *UniAug*: step-size $\gamma$ and regularization strength $\lambda$, and the number of updates in (5) per each sampling step. In addition, inspired by the pseudo labeling strategy (Botao et al., 2023), we provide an option threshold $q$ for the sampling process of the diffusion model. Specifically, we only keep the edges with the probability of existence higher than $q$ for each sampling step. After the sampling process, we recover the training edges of the original graph structure. The hyperparameters are tuned from the choices in Table 14.

Table 14: Hyperparameter choices for link prediction.

| | |
|---|---|
| $\lambda$ | [0.01, 1, 100] |
| $\gamma$ | [0.1, 1.0, 10.0] |
| $q$ | [None, 0.9, 0.99, 0.999] |
| # updates | [5, 10, 20] |

One thing to mention is that we handle the large graphs by graph partitioning with METIS (Karypis & Kumar, 1998). Specifically, we augment the partitions of a large graph and then assemble the partitions back into a single graph. The edges between different partitions are recovered after the assembling process.

Table 15: Effects of different guidance objectives.

| | Cora MRR | Citeseer MRR | Pubmed MRR | Power Hits@10 | Yeast Hits@10 | Erdos Hits@10 | Flickr Hits@10 |
|---|---|---|---|---|---|---|---|
| Link guide | 30.45 ± 2.90 | **54.66 ± 4.55** | 16.97 ± 0.92 | 33.41 ± 2.95 | 25.80 ± 4.10 | 36.79 ± 1.98 | **9.46 ± 1.18** |
| Degree guide | 32.73 ± 6.71 | 51.13 ± 5.51 | 16.37 ± 0.58 | 32.88 ± 2.02 | **27.52 ± 4.80** | **39.67 ± 4.51** | 9.11 ± 0.88 |
| CN guide | **35.36 ± 7.88** | 50.86 ± 5.73 | **17.28 ± 1.89** | **34.36 ± 1.68** | 26.67 ± 4.02 | 36.18 ± 4.32 | 9.28 ± 1.18 |

As mentioned in Section 4.1, we choose three guidance objectives for link prediction with different granularity. The effects of different objectives can be found in Table 15. We observe that the outcomes of different objectives differ across datasets and there is no consistently winning strategy.

## D.3 NODE CLASSIFICATION

For node classification on heterophilic graphs, we use the fixed splits from Geom-GCN (Pei et al., 2020) for Cornell, Wisconsin, Texas, and Actor. For Chameleon and Squirrel, we remove duplicated nodes following (Platonov et al., 2023) and take their fixed splits. Regarding node classification on homophilic graphs, we employ the semi-supervised setting (Yang et al., 2016). The GCN backbone is implemented as a 2-layer classifier. Similar to graph property prediction, we have mainly four hyperparameters for *UniAug*: step-size $\gamma$ and regularization strength $\lambda$ in (5), number of repeats per training graph, and whether augment validation and test graphs with the trained guidance head. The hyperparameters are tuned from the choices in Table 16.

Table 16: Hyperparameter choices for node classification.

| | |
|---|---|
| $\lambda$ | 0.01 |
| $\gamma$ | [0.1, 0.5, 1.0] |
| # repeats | [1, 5, 10] |
| Aug val and test | [True, False] |

## D.4 INVESTIGATION ON SCALING

In Section 4.2, we investigate the scaling behavior of *UniAug* regarding data scale and pre-training time. We omit some of the results for a better visualization. Here we present the numerical results in Table 17 and Table 18.

Table 17: Effects of pre-training data scale on graph classification (up) and link prediction (down).

| | Enzymes | Proteins | IMDB-B | IMDB-M |
|---|---|---|---|---|
| GIN | 66.00 ± 7.52 | 73.32 ± 4.03 | 71.10 ± 2.90 | 49.07 ± 2.81 |
| *UniAug*- SMALL | 66.83 ± 7.38 | 73.50 ± 5.61 | 69.80 ± 2.70 | 48.93 ± 3.20 |
| *UniAug*- FULL | **71.33 ± 6.51** | 74.05 ± 4.82 | 73.11 ± 2.35 | 49.67 ± 2.41 |
| *UniAug*- EXTRA | 71.17 ± 7.10 | **75.47 ± 2.50** | **73.50 ± 2.48** | **50.13 ± 2.05** |

| | Cora MRR | Citeseer MRR | Power Hits@10 | Yeast Hits@10 | Erdos Hits@10 |
|---|---|---|---|---|---|
| GCN | 30.26 ± 4.80 | 50.57 ± 7.91 | 30.61 ± 4.07 | 24.71 ± 4.92 | 35.71 ± 2.65 |
| *UniAug*- SMALL | 32.25 ± 8.71 | 47.91 ± 3.87 | 32.25 ± 3.72 | 25.81 ± 4.89 | 36.28 ± 3.56 |
| *UniAug*- FULL | 32.81 ± 7.44 | 48.32 ± 6.00 | 32.97 ± 3.75 | 26.36 ± 4.62 | 36.07 ± 4.20 |
| *UniAug*- EXTRA | **35.36 ± 7.88** | **54.66 ± 4.55** | **34.36 ± 1.68** | **27.52 ± 4.80** | **39.67 ± 4.51** |

Table 18: Effects of pre-training amount of compute on graph classification (up) and link prediction (down).

| $10^{-3}$ PF-days | Enzymes | Proteins | IMDB-B | IMDB-M |
|---|---|---|---|---|
| 5 | 68.18 ± 6.21 | 73.32 ± 3.63 | 71.20 ± 2.90 | 48.28 ± 2.75 |
| 10 | 69.00 ± 5.10 | 74.30 ± 5.33 | 72.80 ± 3.85 | 48.60 ± 2.23 |
| 15 | 68.83 ± 5.88 | 75.11 ± 3.18 | 71.77 ± 2.38 | 48.60 ± 2.48 |
| 20 | 70.79 ± 5.73 | 74.87 ± 5.30 | 73.04 ± 2.82 | 49.47 ± 2.20 |
| 25 | **71.50 ± 5.85** | **75.47 ± 2.50** | **73.50 ± 2.48** | **50.13 ± 2.05** |

| $10^{-3}$ PF-days | Cora MRR | Citeseer MRR | Power Hits@10 | Yeast Hits@10 | Erdos Hits@10 |
|---|---|---|---|---|---|
| 5 | 27.56 ± 4.36 | 49.45 ± 9.20 | 22.81 ± 9.47 | 23.62 ± 9.77 | 35.33 ± 3.16 |
| 10 | 31.02 ± 6.53 | 50.72 ± 6.22 | 32.49 ± 2.52 | 26.70 ± 4.85 | 36.10 ± 4.66 |
| 15 | 33.24 ± 7.97 | 49.02 ± 5.92 | 32.88 ± 3.31 | **27.80 ± 4.55** | **39.70 ± 3.67** |
| 20 | 34.71 ± 9.08 | 52.90 ± 3.84 | 33.69 ± 3.23 | 26.90 ± 3.93 | 39.33 ± 3.16 |
| 25 | **35.36 ± 7.88** | **54.66 ± 4.55** | **34.36 ± 1.68** | 27.52 ± 4.80 | 39.67 ± 4.51 |

## E BROADER IMPACT

In this work, we build a universal graph structure augmentor that benefits from data scaling across domains. Given the consistent performance improvements for different tasks, we expect this work to contribute significantly towards the goal of building a graph foundation model. In the meantime, we showcase the power of the deep generative models on graphs by introducing new application scenarios. We anticipate such success will contribute to the community of generative models and graph learning.

It is important to mention that the model backbones of our method and baselines heavily rely on neighboring node information as an inductive bias. However, this characteristic can result in biased predictions, especially when patterns in neighborhood majorities dominate, leading to potential ethical issues in model predictions.

