# OpenReview forum: "Cross-Domain Graph Data Scaling: A Showcase with Diffusion Models"
_ICLR.cc/2025/Conference — Submitted to ICLR 2025_

### Official Review · Reviewer_y1X3 · 2024-10-27

**Soundness:** 2
**Presentation:** 3
**Contribution:** 2
**Rating:** 5
**Confidence:** 4

**Summary:**

The paper proposes a model called UniAug, based on a graph diffusion model. The approach consists of two steps: first, pretraining on diverse graphs from different domains, and second, using the model to generate synthetic structures for downstream tasks. Experimental results on approximately 30 cross-domain graphs demonstrate the empirical effectiveness of the method.

**Strengths:**

1. The paper is well-written and easy to follow, presenting a simple yet effective approach that uses diffusion models to augment structural data for cross-domain graphs.

2. Addressing the challenge of designing models for cross-graph learning is highly valuable and relevant.

3. The experiments are comprehensive, covering approximately 30 graphs across more than five domains.

**Weaknesses:**

1. The authors claim to propose a graph foundation model for cross-domain graphs. However, the method appears to be more of a universal graph structural augmentor that utilizes graph diffusion models for generating additional structures, rather than a foundation model capable of inference across various graph types, like [1,2,3]. Additionally, although the authors demonstrate that pretrained diffusion models can generate synthetic structures to enhance performance on various downstream tasks, the concept of using graph diffusion models to create structures is not entirely new [4,5], which may limit the novelty of the approach.

2. The experimental results are not entirely convincing. The authors use node degrees as features for self-supervised baselines, which can significantly impair model performance when replacing original node features with node degrees, whereas using original node features for the proposed method. Although the authors elaborate on this issue in Appendix D, the experimental descriptions still confuse me, and I cannot find strong evidence that the proposed UniAug outperforms existing SSL methods under comparable settings (i.e., using the same feature setup).

3. The applicability of the method may be limited in certain scenarios. When applying it to downstream tasks, such as node and graph classification, the authors use node labels to guide the generation of graph structures. However, the method may struggle with graphs that lack sufficient label information. It would be helpful to understand whether the approach can be effectively applied to downstream graphs with limited or no label data.

References:

[1] One for all: Towards training one graph model for all classification tasks. ICLR, 2024.

[2] GraphAny: A Foundation Model for Node Classification on Any Graph. Arxiv, 2024.

[3] Zerog: Investigating cross-dataset zero-shot transferability in graphs. KDD, 2024.

[4] SimDiff: Simple Denoising Probabilistic Latent Diffusion Model for Data Augmentation on Multi-modal Knowledge Graph. KDD, 2024.

[5] Data-Centric Learning from Unlabeled Graphs with Diffusion Model. NeurIPS, 2023.

**Questions:**

1. The authors present experimental results with varying pretraining scales in Figure 3. Although there is a general performance improvement between the SMALL, FULL, and EXTRA scales, there are instances where the model pretrained on smaller graph datasets outperforms those pretrained on larger datasets, such as on the Enzymes and Erdos datasets. Could the authors provide a detailed explanation for these observations?

2. Can the proposed method be applied to more practical scenarios, such as few-shot or zero-shot learning?

3. The proposed method leverages a diffusion model to generate additional structural information. While the model demonstrates empirically desirable performance, I am curious whether there is a theoretical understanding that supports its efficacy.

---

> ### Author Response · Authors · 2024-11-18
> **Response to reviewer y1X3 (1/3)**
>
> > Q1. The authors claim to propose a graph foundation model for cross-domain graphs. However, the method appears to be more of a universal graph structural augmentor rather than a foundation model. Additionally, the concept of using graph diffusion models to create structures is not entirely new, which may limit the novelty of the approach.
>
> We agree that UniAug functions as a universal graph structural augmentor, as stated in line 18 of our manuscript:
>
> - We propose UniAug, a universal graph structure augmentor built on a diffusion model.
>
> We emphasize that UniAug is a universal method capable of leveraging graphs from various domains and is applicable to a wide range of downstream tasks. Furthermore, we want to highlight UniAug’s flexibility compared to the models mentioned by the reviewer. Specifically, both GraphAny and Zerog are designed exclusively for node classification, limiting their application scenarios. In contrast, UniAug’s data augmentation paradigm allows it to be applied to any desired downstream task, including but not limited to node classification, link prediction, and graph classification. According to TSGFM [1], the “one-for-all” approach leads to performance drops for node classification when compared to the GCN baseline. However, UniAug consistently provides performance improvements over vanilla GNN baselines across various tasks, including node classification, link prediction, and graph classification.
>
> One of the key contributions of our work is **cross-domain data scaling**. Existing approaches to graph data augmentation and graph pre-training are restricted to in-domain datasets or rely on handcrafted data filtering strategies, which utilize only a small fraction of publicly available data. UniAug leverages a diffusion model to learn graph data distributions across domains and subsequently performs structure augmentations on downstream datasets. Our experimental results demonstrate that UniAug not only achieves cross-domain data scaling but also delivers benefits across diverse downstream tasks, regardless of dataset scale or domain.
>
> > Q2. The authors use node degrees as features for self-supervised baselines, which can significantly impair model performance. Although the authors elaborate on this issue in Appendix D, the experimental descriptions still confuse me, and I cannot find strong evidence that the proposed UniAug outperforms existing SSL methods under comparable settings
>
> In our manuscript, we pre-train all self-supervised baselines on the same pre-training set as UniAug to benchmark performance in a multi-domain pre-training setting. As the pre-training data coverage increases, we encounter diverse feature heterogeneity challenges, including **missing features** and **mismatched semantics**. While existing baselines are not well-suited to handle missing feature issues, we address this limitation by calculating node degrees as proxy node features for these baselines.
>
> We acknowledge that this workaround may contribute to performance drops for the baselines. To provide a more comprehensive evaluation, we include both semi-supervised and self-supervised results for graph classification in Appendix D. In the semi-supervised setting, for each data split, the baseline is pre-trained on the train and test sets in a self-supervised manner and then fine-tuned on the training set with labels. In the self-supervised setting, the baselines are pre-trained on the entire dataset, and the learned graph embeddings are subsequently classified using a downstream SVM classifier.
>
> ***
> *To be continued in the next reply*

---

> ### Author Response · Authors · 2024-11-18
> **Response to reviewer y1X3 (2/3)**
>
> To further illustrate the performance improvements of UniAug compared to existing SSL methods, we conducted two additional experiments:
>
> 1. **Comparison with self-supervised baselines**: We compare the performance of UniAug with four representative self-supervised baselines on node classification and link prediction, as summarized in the following tables. The best results are highlighted in bold, with the last column indicating the average ranking. Our results demonstrate that UniAug consistently outperforms the baselines in terms of average ranking, particularly on heterophilic node classification datasets.
>
> |  |  | Cora | Citeseer | Pubmed | Power | Yeast | Erdos | Flickr |  |
> | --- | --- | --- | --- | --- | --- | --- | --- | --- | --- |
> |  |  | MRR | MRR | MRR | Hits@10 | Hits@10 | Hits@10 | Hits@10 | A.R. |
> | Self-supervised | MVGRL[1] | 29.13±3.90 | 51.32±4.12 | 15.21±2.35 | 31.71±3.78 | 23.74±5.74 | 36.21±2.81 | 8.42±2.18 | 4.29 |
> |  | GRACE[2] | 31.77±4.31 | 49.13±3.95 | 16.88±1.74 | 28.21±5.04 | 23.96±4.31 | 33.90±2.12 | **9.87±0.98** | 4.00 |
> |  | BGRL[3] | 33.59±2.14 | 51.91±5.01 | 16.93±2.03 | 33.71±3.21 | 25.91±3.12 | 37.95±1.73 | 8.52±1.85 | 2.57 |
> |  | GraphMAE[4] | 32.98±5.01 | 52.71±5.39 | **18.83±1.30** | 32.81±2.12 | 26.51±2.92 | 35.63±3.61 | 7.01±3.86 | 2.86 |
> | GDA | UniAug | **35.36±7.88** | **54.66±4.55** | 17.28±1.89 | **34.36±1.68** | **27.52±4.80** | **39.67±4.51** | 9.46±1.18 | 1.29 |
>
> | ACC ↑ |  | Cornell | Wisconsin | Texas | Actor | Chameleon* | Squirrel* | A.R. |
> | --- | --- | --- | --- | --- | --- | --- | --- | --- |
> | Self-supervised | MVGRL[1] | 56.19±2.42 | 50.64±5.89 | 61.70±3.94 | 31.37±0.83 | 32.34±2.11 | 35.32±1.32 | 4.00 |
> |  | GRACE[2] | 56.39±2.11 | 53.83±3.56 | 63.54±2.57 | 28.14±0.81 | 35.71±1.95 | 33.65±2.51 | 4.17 |
> |  | BGRL[3] | 56.67±2.13 | 59.80±4.08 | 65.78±2.66 | 29.80±0.31 | 37.01±2.89 | 34.77±2.01 | 2.67 |
> |  | GraphMAE[4] | 57.31±2.11 | 58.27±2.91 | 58.34±3.57 | 28.97±0.27 | 36.75±1.78 | 39.13±2.01 | 3.17 |
> | GDA | UniAug | **68.11±6.72** | **69.02±4.96** | **73.51±5.06** | **33.11±1.57** | **43.84±3.39** | **41.90±1.90** | 1.00 |
>
> 2. **Few-shot graph classification**: In response to the reviewer’s suggestion, we include results for 5-shot graph classification following [6]. These results show significant performance improvements of UniAug over the self-supervised baselines, highlighting the effectiveness of our method under few-shot settings.
>
> | 5-shot ACC ↑ | Proteins | Enzymes |
> | --- | --- | --- |
> | GIN | 58.17±8.58 | 20.34±5.01 |
> | InfoGraph | 54.12±8.20 | 20.90±3.32 |
> | GraphCL | 56.38±7.24 | 28.11±4.00 |
> | JOAO | 57.21±6.91 | 35.31±3.79 |
> | GraphMAE | 58.03±5.35 | 33.91±6.58 |
> | UniAug | **66.85±4.71** | **48.37±4.77** |
>
> > Q3. When applying it to downstream tasks, such as node and graph classification, the authors use node labels to guide the generation of graph structures. However, the method may struggle with graphs that lack sufficient label information. It would be helpful to understand whether the approach can be effectively applied to downstream graphs with limited or no label data.
>
> In Section 3.3, we described various guidance objectives at the node, edge, and graph levels, including both supervised and self-supervised objectives. To demonstrate that UniAug can still achieve comparable performance without label guidance, we conducted experiments using node degree as the guidance objective on graph classification datasets. The results are presented in the following table, where we observe that UniAug, guided by node degree, consistently outperforms the GIN baseline across all datasets.
>
> | ACC ↑ | DD | Proteins | NCI1 | IMDB-B | IMDB-M | Reddit-B | Collab |
> | --- | --- | --- | --- | --- | --- | --- | --- |
> | GIN | 75.81 ± 6.11 | 73.32 ± 4.03 | 78.30 ± 3.20 | 71.10 ± 2.90 | 49.07 ± 2.81 | 90.85 ± 1.30 | 74.54 ± 2.41 |
> | UniAug - Node Degree | 76.77±2.58 | 76.01±1.77 | 79.33±0.97 | 72.75±1.98 | 50.76±2.36 | 91.85±1.88 | 75.82±1.61 |
> | UniAug - Label | 78.13±2.61 | 75.47±2.50 | 80.54±1.77 | 73.50±2.48 | 50.13±2.05 | 92.28±1.59 | 77.00±2.02 |
>
> Additionally, as mentioned in our response to Q2, we provide 5-shot graph classification results, which show that UniAug significantly outperforms all self-supervised baselines by a large margin. In summary, while we leverage label guidance when labels are available to achieve better performance, UniAug remains effective with limited labels or self-supervised guidance objectives, demonstrating its adaptability.
>
> ***
> *To be continued in the next reply*

---

> ### Author Response · Authors · 2024-11-18
> **Response to reviewer y1X3 (3/3)**
>
> > Q4. The authors present experimental results with varying pretraining scales in Figure 3. Although there is a general performance improvement between the SMALL, FULL, and EXTRA scales, there are instances where the model pretrained on smaller graph datasets outperforms those pretrained on larger datasets, such as on the Enzymes and Erdos datasets. Could the authors provide a detailed explanation for these observations
>
> In the SMALL, FULL, and EXTRA scales, we progressively increase the distribution coverage of the pre-training set, resulting in an overall upward trend in downstream performance. We emphasize that the downstream performance is closely tied to the distribution coverage of the pre-training set, which explains the observed performance fluctuations on the Enzymes and Erdos datasets.
>
> For the Enzymes dataset, we observe a slight performance drop when adding 1000 graphs from the Github Star dataset to the FULL set. Graphs in the Enzymes dataset represent small molecules, which differ significantly from those in the Github Star dataset in terms of both the number of nodes and structural properties. Consequently, it is expected that UniAug trained on the EXTRA set does not substantially outperform the FULL set for this dataset. However, it is important to note that the addition of these 1000 graphs results in performance improvements across all other datasets, with only a negligible drop in performance on Enzymes.
>
> Regarding the Erdos dataset, its distribution occupies the scattered space illustrated in Fig. 2, with minimal overlap with graphs in the Network Repository. This limited overlap explains why we do not observe significant performance gains when training UniAug on the SMALL or FULL sets. However, in the EXTRA set, the inclusion of 1000 additional graphs effectively fills the scattered space, leading to a substantial performance boost on the Erdos dataset.
>
> > Q5. Can the proposed method be applied to more practical scenarios, such as few-shot or zero-shot learning?
>
> Please see the answer to Q2.
>
> > Q6. While the model demonstrates empirically desirable performance, I am curious whether there is a theoretical understanding that supports its efficacy.
>
> One of the key motivations of this manuscript is that GNN model architectures should be aligned with task-specific inductive biases [7]. For instance, GCN performs well on node classification for homophilic graphs, while GIN demonstrates superior performance on graph classification. This highlights the inherent difficulty in designing a single GNN that performs well across all types of tasks.
>
> To address this, graph data augmentation (GDA) methods aim to mitigate the issue by directly modifying the data, effectively aligning data patterns with model designs. For example, Half-Hop [8] adds nodes on each edge to improve GCN's performance on node classification tasks for heterophilic graphs, achieving substantial gains. CFLP [9] generates counterfactual links to enhance link prediction. However, while GDA methods offer performance improvements, they often rely on handcrafted augmentation strategies tailored for specific tasks or datasets, limiting their generalizability to broader applications.
>
> UniAug addresses this issue by leveraging a diffusion model. Intuitively, UniAug learns diverse data patterns by pre-training on graphs across domains, enabling it to adaptively augment downstream graphs with task-specific and data-specific guidance. This allows UniAug to "automatically" align data patterns with the downstream GNN architecture, resulting in empirical success across a wide range of tasks and datasets.
>
> [1] Chen, Zhikai, et al. “Text-space Graph Foundation Models: Comprehensive Benchmarks and New Insights.” NeurIPS, 2024
>
> [2] Hassani, Kaveh, and Amir Hosein Khasahmadi. "Contrastive multi-view representation learning on graphs." ICML, 2020.
>
> [3] Zhu, Yanqiao, et al. "Deep graph contrastive representation learning." arXiv, 2020.
>
> [4] Thakoor, Shantanu, et al. "Large-Scale Representation Learning on Graphs via Bootstrapping." ICLR, 2022
>
> [5] Hou, Zhenyu, et al. "Graphmae: Self-supervised masked graph autoencoders." KDD, 2022.
>
> [6] Liu, Zemin, et al. "Graphprompt: Unifying pre-training and downstream tasks for graph neural networks." WWW, 2023.
>
> [7] Mao, Haitao, et al. "Demystifying structural disparity in graph neural networks: Can one size fit all?." NeurIPS, 2024
>
> [8] Azabou, Mehdi, et al. "Half-Hop: A graph upsampling approach for slowing down message passing." ICML, 2023.
>
> [9] Zhao, Tong, et al. "Learning from counterfactual links for link prediction." ICML, 2022.
>
> ***
> Thank you again for your constructive feedback and suggestions, which have greatly contributed to improving our work. We believe the additional results and explanations address your concerns and provide a clearer picture of the contributions of our study. We welcome any further input and hope for your strong support for the paper, reflected in an improved score.

---

> ### Author Response · Authors · 2024-12-02
> **Gentle Reminder**
>
> Thank you for your thoughtful and constructive review. We sincerely appreciate the time and effort you’ve invested in providing such valuable feedback. If any questions or concerns remain, we would be happy to continue the discussion and provide further clarification. We look forward to your reevaluation of our work.
>
> Best regards, The Authors

---

### Official Review · Reviewer_sW2Z · 2024-10-29

**Soundness:** 3
**Presentation:** 3
**Contribution:** 3
**Rating:** 6
**Confidence:** 4

**Summary:**

This paper explores a meaningful issue: How to effectively leverage the increasing scale of data across domains for graph learning? To achieve effective data scaling, this paper proposes a universal graph augmentation framework, UniAug. This framework pre-trains a discrete diffusion model on massive graphs across domains to learn the structural patterns, and conducts structure augmentation with the help of the pre-trained diffusion model. Experiments shows that this framework can leverage the data scaling laws and achieve performance improvements across various downstream tasks.

**Strengths:**

+ The research topic of this paper—How to effectively leverage the increasing scale of data across domains for graph learning, is highly significant and meaningful, representing a critical issue that urgently needs to be addressed in the current field.
+ This paper collects thousands of graphs from varied domains with diverse patterns to explore the potential of data scaling for graph learning.
+ This paper provides a solution based on the diffusion model from a structural augmentation perspective. This augmentation paradigm strategically circumvents feature heterogeneity and fully utilizes downstream inductive biases in a plug-and-play manner.

**Weaknesses:**

+ The authors should clearly illustrate the scale and domain variety of the pre-training data collection in the paper and make it public for the community to conduct further research on the issue. Meanwhile, providing code examples is also essential.
+ The authors should consider whether excessively introducing additional datasets for pre-training is appropriate and whether it might produce negative effects.
+ Related work about data scaling on graphs should be include in the paper [1, 2, 3].
1. [1] Liu J, Mao H, Chen Z, et al. Neural scaling laws on graphs[J]. arXiv preprint arXiv:2402.02054, 2024.
2. [2] Ma Q, Mao H, Liu J, et al. Do Neural Scaling Laws Exist on Graph Self-Supervised Learning?[J]. arXiv preprint arXiv:2408.11243, 2024.
3. [3] Wang Z, Li Y, Ding B, et al. Exploring Neural Scaling Law and Data Pruning Methods For Node Classification on Large-scale Graphs[C]//Proceedings of the ACM on Web Conference 2024. 2024: 780-791.

**Questions:**

See Weaknesses.

---

> ### Author Response · Authors · 2024-11-18
> **Response to reviewer sW2Z**
>
> > Q1. The authors should clearly illustrate the scale and domain variety of the pre-training data collection in the paper and make it public for the community to conduct further research on the issue. Meanwhile, providing code examples is also essential.
>
> We thank the reviewer for their suggestions regarding data and code. Below, we provide more details about our pre-training data collection process.
>
> Our pre-training dataset is primarily based on the Network Repository, which contains thousands of graphs spanning 33 categories, including biological networks, social networks, and others. We collected all available graphs from the Network Repository, removed duplicates, and filtered out outliers as described in Section 3.1. This process resulted in a collection of 3000 graphs. However, as noted in Section 3.1, the distribution coverage of these graphs was incomplete. To address this, we incorporated an additional 1000 graphs from the Github Star dataset, forming our complete pre-training collection. Both the Network Repository and Github Star datasets are publicly accessible, and we will make the data processing code available upon acceptance of the manuscript.
>
> Regarding the source code for the experiments, we have included it in the supplementary files. The submission also contains running scripts to reproduce the experiments. Upon acceptance, we will provide detailed tutorials to ensure reproducibility and facilitate the use of our methods.
>
> > Q2. The authors should consider whether excessively introducing additional datasets for pre-training is appropriate and whether it might produce negative effects.
>
> We thank the reviewer for their valuable suggestion regarding data. We would like to emphasize the importance of data quality in our approach. As discussed in Section 3.1, we filtered out certain outliers from the Network Repository. Specifically, there are extreme cases such as fully connected graphs and empty graphs (i.e., graphs without edges), which differ significantly from the majority of graphs. To ensure the model learns meaningful distributions, we excluded these extreme cases from our pre-training dataset.
>
> Another critical factor in data scaling is the self-labeling process described in Section 3.2. Our pre-training dataset comprises graphs with diverse patterns. To enable the diffusion model to effectively distinguish these patterns, we incorporated a self-supervised labeling process and integrated the self-assigned labels into the model. This step is crucial for achieving positive transfer, as evidenced by the significant performance drops observed in Table 7 when the self-labeling process is omitted.
>
> Finally, we refer to the experiments conducted with three pre-training datasets: SMALL, FULL, and EXTRA collections. As we increase both the scale and diversity of the data from SMALL to EXTRA, we observe a clear trend of performance improvement, as shown in Fig. 3. These findings highlight that, with proper outlier removal and the inclusion of the self-labeling process, our diffusion model and data augmentation paradigm can effectively scale to larger and more diverse datasets.
>
> > Q3. Related work about data scaling on graphs should be included in the paper.
>
> We thank the reviewer for suggesting the related works. We will include the mentioned references in the revised manuscript.
>
> ***
> We greatly appreciate the valuable feedback and suggestions provided, which have helped us strengthen the manuscript. We sincerely appreciate your thoughtful feedback and look forward to your further input and support for the paper.

---

> ### Comment · Reviewer_sW2Z · 2024-11-26
>
> Thank you for the authors' rebuttal. However, I still have the following concerns.
>
> First, for the datasets and codes released after the acceptance of the paper, it seems to be a less convincing promise, which significantly reduces the reproducibility of  the paper. I will not buy the clarification.
>
> Second, I am curious about how to ensure the accuracy of labels output by the self-supervised labeling process.
>
> Third, the authors do not reply to our second question: whether excessively introducing additional datasets for pre-training is appropriate?

---

> ### Author Response · Authors · 2024-11-27
> **Thank you for the reply**
>
> We thank the reviewer for the follow-up questions and concerns. Below, we address each point in detail:
> > Q1. For the datasets and codes released after the acceptance of the paper, it seems to be a less convincing promise, which significantly reduces the reproducibility of the paper.
>
> As mentioned in our previous response, we have included all necessary resources for reproducibility in the **supplementary material** (a zip file) of the submission. This includes:
>
> - Code for data downloading and preprocessing,
> - Processed pre-training data,
> - Pre-trained model weights, and
> - Running scripts.
>
> The provided bash scripts in the `run` directory enable full reproduction of the results. We believe this ensures a high level of reproducibility and addresses concerns about code and dataset availability.
>
> > Q2. How to ensure the accuracy of labels output by the self-supervised labeling process.
>
> The self-supervised labeling process functions as a clustering mechanism based on graph properties, and as such, does not have a notion of "accuracy" in the traditional supervised learning sense. The primary purpose of this process is to identify and group distribution patterns of the graphs, which the diffusion model leverages to enhance its understanding of diverse graph patterns.
>
> > Q3. Whether excessively introducing additional datasets for pre-training is appropriate?
>
> As noted in our earlier response, increasing both the scale and diversity of datasets for pre-training has consistently resulted in performance improvements across various tasks and datasets. **From a performance perspective, this demonstrates that the inclusion of additional datasets for pre-training is not only appropriate but also beneficial**.

---

> ### Author Response · Authors · 2024-12-02
> **Gentle Reminder**
>
> Thank you for your insightful review and for dedicating your time to providing valuable feedback. Based on your comments, we have clarified our statements to best resolve your concerns about our manuscript. We would greatly appreciate it if you could share any further questions or comments.
>
> Best regards, The Authors

---

### Official Review · Reviewer_9F3c · 2024-11-04

**Soundness:** 2
**Presentation:** 2
**Contribution:** 2
**Rating:** 3
**Confidence:** 3

**Summary:**

The paper proposes a graph structure augmentation framework built on discrete diffusion models. They first pre-train a structure-only discrete diffusion model on graphs from multiple domains. During downstream adaptation, the model generates synthetic graph through guided generation, using a task-specific head to align these generated structures with the target task. This framework aims to enhance performance across various tasks, including node classification, link prediction and graph prediction. The experiment also shows that the downstream performance increases when the amount of pre-training data increases.

**Strengths:**

- The paper demonstrates an interesting cross-domain transfer ability, where the pre-trained model can be adapted to various downstream tasks at different levels (node, edge, and graph).
- The presentation is clear, with well-explained methods and experimental results, making it easy to follow the proposed approach and findings.

**Weaknesses:**

- I have some concerns about the comparison of UniAug with baseline pre-training methods. The authors mention that they used the same pre-training dataset for UniAug and baselines. They also calculated node degrees as inputs and replaced node features with node degrees for downstream testing. However, these baseline models were not originally designed to be trained and tested in this way. As shown in Table 13, when evaluated in their original semi-supervised or self-supervised settings, these baselines achieve results similar to UniAug. This is different from the larger improvement suggested in Table 2. A more accurate comparison would keep the baselines in their original configurations, and it would also help to see similar comparisons for link prediction and node classification tasks.
- Additionally, a discussion of pre-training time for UniAug compared to the baselines would improve the analysis. This can offer a better understanding of UniAug’s efficiency compared to other methods.
- A more detailed description of the sizes of Small, Full, and Extra pre-training datasets should be provided. From the description, the Extra dataset includes 1,000 more subgraphs from the GitHub Star dataset than the Full dataset. This addition results in large improvements for link prediction tasks but shows little effect on graph classification tasks. Thus I am curious whether the improvements come from the increased amount of data or from the diversity added by the new subgraphs. A more detailed breakdown of the dataset sizes and an analysis of data diversity versus quantity would help clarify the reasons behind these improvements.

**Questions:**

Please see the weakness part.

---

> ### Author Response · Authors · 2024-11-18
> **Response to reviewer 9F3c (1/2)**
>
> > Q1. When evaluated in their original semi-supervised or self-supervised settings, these baselines achieve results similar to UniAug. A more accurate comparison would keep the baselines in their original configurations, and it would also help to see similar comparisons for link prediction and node classification tasks.
>
> In our manuscript, we pre-train all self-supervised baselines on the same pre-training set as UniAug to benchmark performance in a multi-domain pre-training setting. As the pre-training data coverage increases, we encounter diverse feature heterogeneity challenges, including **missing features** and **mismatched semantics**. While existing baselines are not well-equipped to handle missing feature issues, we address this limitation by calculating node degrees as node features for these baselines.
>
> We acknowledge that this workaround may lead to performance drops for the baselines. To provide a more comprehensive evaluation, we include both semi-supervised and self-supervised results for graph classification in Appendix D. Additionally, we present self-supervised results for link prediction and node classification in the following tables. The best results are highlighted in bold, with the last column showing the average ranking. Our results demonstrate that UniAug consistently outperforms the baselines in terms of average ranking, particularly on heterophilic node classification datasets.
>
> |  |  | Cora | Citeseer | Pubmed | Power | Yeast | Erdos | Flickr |  |
> | --- | --- | --- | --- | --- | --- | --- | --- | --- | --- |
> |  |  | MRR | MRR | MRR | Hits@10 | Hits@10 | Hits@10 | Hits@10 | A.R. |
> | Self-supervised | MVGRL[1] | 29.13±3.90 | 51.32±4.12 | 15.21±2.35 | 31.71±3.78 | 23.74±5.74 | 36.21±2.81 | 8.42±2.18 | 4.29 |
> |  | GRACE[2] | 31.77±4.31 | 49.13±3.95 | 16.88±1.74 | 28.21±5.04 | 23.96±4.31 | 33.90±2.12 | **9.87±0.98** | 4.00 |
> |  | BGRL[3] | 33.59±2.14 | 51.91±5.01 | 16.93±2.03 | 33.71±3.21 | 25.91±3.12 | 37.95±1.73 | 8.52±1.85 | 2.57 |
> |  | GraphMAE[4] | 32.98±5.01 | 52.71±5.39 | **18.83±1.30** | 32.81±2.12 | 26.51±2.92 | 35.63±3.61 | 7.01±3.86 | 2.86 |
> | GDA | UniAug | **35.36±7.88** | **54.66±4.55** | 17.28±1.89 | **34.36±1.68** | **27.52±4.80** | **39.67±4.51** | 9.46±1.18 | 1.29 |
>
> | ACC ↑ |  | Cornell | Wisconsin | Texas | Actor | Chameleon* | Squirrel* | A.R. |
> | --- | --- | --- | --- | --- | --- | --- | --- | --- |
> | Self-supervised | MVGRL[1] | 56.19±2.42 | 50.64±5.89 | 61.70±3.94 | 31.37±0.83 | 32.34±2.11 | 35.32±1.32 | 4.00 |
> |  | GRACE[2] | 56.39±2.11 | 53.83±3.56 | 63.54±2.57 | 28.14±0.81 | 35.71±1.95 | 33.65±2.51 | 4.17 |
> |  | BGRL[3] | 56.67±2.13 | 59.80±4.08 | 65.78±2.66 | 29.80±0.31 | 37.01±2.89 | 34.77±2.01 | 2.67 |
> |  | GraphMAE[4] | 57.31±2.11 | 58.27±2.91 | 58.34±3.57 | 28.97±0.27 | 36.75±1.78 | 39.13±2.01 | 3.17 |
> | GDA | UniAug | **68.11±6.72** | **69.02±4.96** | **73.51±5.06** | **33.11±1.57** | **43.84±3.39** | **41.90±1.90** | 1.00 |
>
> In summary, UniAug stands out over the baselines in two key aspects:
>
> 1. **Versatility:** The structure-only diffusion model of UniAug enables pre-training across graphs from different domains, even in cases where most graphs lack corresponding node features. Furthermore, UniAug’s data augmentation paradigm makes it adaptable to various graph-related tasks, including but not limited to node classification, link prediction, and graph classification.
> 2. **Performance:** As demonstrated in the manuscript and supported by the tables above, UniAug achieves superior performance compared to the baselines in both the multi-domain pre-training and self-supervised settings.
>
> [1] Hassani, Kaveh, and Amir Hosein Khasahmadi. "Contrastive multi-view representation learning on graphs." ICML, 2020.
>
> [2] Zhu, Yanqiao, et al. "Deep graph contrastive representation learning." arXiv, 2020.
>
> [3] Thakoor, Shantanu, et al. "Large-Scale Representation Learning on Graphs via Bootstrapping." ICLR, 2022
>
> [4] Hou, Zhenyu, et al. "Graphmae: Self-supervised masked graph autoencoders." KDD, 2022.
>
> ***
> *To be continued in the next reply*

---

> > ### Author Response · Authors · 2024-11-18
> > **Response to reviewer 9F3c (2/2)**
> >
> > > Q2. A discussion of pre-training time for UniAug compared to the baselines would improve the analysis
> >
> > We thank the reviewer for the valuable suggestion. Below, we summarize the pre-training time comparison in a table. UniAug requires less time per epoch and benefits significantly from additional training epochs due to the nature of diffusion models.
> >
> > |  | # sec per epoch | # epochs |
> > | --- | --- | --- |
> > | UniAug | 10 | 10000 |
> > | JOAO | 35 | 1000 |
> > | D-SLA | 33 | 1000 |
> > | GraphMAE | 24 | 1000 |
> >
> > It is important to highlight that this comparison is conducted under the **multi-domain pre-training setting**, where **most baselines experience a performance decline compared to vanilla GNNs**. In contrast, UniAug consistently delivers performance improvements across various tasks and datasets. While the inclusion of the diffusion model necessitates more training iterations, it enables UniAug to achieve a better scaling effect when pre-training on graphs from diverse domains, making it highly effective for tasks with varying levels of granularity.
> >
> > > Q3. A more detailed description of the sizes of Small, Full, and Extra pre-training datasets should be provided. A more detailed breakdown of the dataset sizes and an analysis of data diversity versus quantity would help clarify the reasons behind these improvements
> >
> > We thank the reviewer for pointing out the issues in the data scaling experiments. Below, we provide a more detailed explanation of the three pre-training sets and present additional data scaling results.
> >
> > - **SMALL:** The Network Repository contains graphs from 33 different categories. For the SMALL set, we randomly sampled 10 graphs per category.
> > - **FULL:** We collected all graphs from the Network Repository, removed duplicates, and filtered out outliers as described in Section 3.1. This resulted in a collection of 3000 graphs, which we refer to as the FULL set.
> > - **EXTRA:** Noticing the incomplete distribution coverage of the FULL set (Section 3.1), we incorporated 1000 additional graphs from the Github Star dataset, forming the EXTRA set.
> >
> > We recognize that these three sets vary in both scale and diversity. To analyze the scaling effect of UniAug based solely on data quantity, we clustered the pre-training set into 10 clusters based on graph-level representations (Section 3.2) and performed stratified sampling within these clusters. From this, we created three subsets containing 25%, 50%, and 75% of the total graphs, and pre-trained UniAug on each subset.
> >
> > The results, summarized in the following figures with links for [graph classification](https://drive.google.com/file/d/15wiJj9awBqxYbvJirY99O3RLwZsUR8pv/view?usp=sharing) and [link prediction](https://drive.google.com/file/d/19WicL43j9YEwWHbXZsBI9b93OjOTznJl/view?usp=sharing), show a clear trend of performance improvement as the size of the pre-training set increases. Combined with experiments on the SMALL, FULL, and EXTRA sets, these findings suggest that UniAug benefits from both increasing the scale and enhancing the diversity of the pre-training data.
> >
> > ***
> >
> > We hope that the additional experiments, discussions, and clarifications lead to a better appreciation of the contributions of our work. We hope that we have fully addressed your concerns, and we are looking forward to your further input and your support of the paper in score.

---

> > > ### Comment · Reviewer_9F3c · 2024-11-19
> > >
> > > Thank you for your detailed response and the additional experiment. While I appreciate the effort, I still have some concerns.
> > > - Q1. The model shows strong performance on heterophilic node classification, but the baselines you compare with are designed for homophilic data. It would be better to include results on homophilic datasets like Cora and PubMed to provide a balanced comparison. To fairly evaluate performance in heterophilic settings, you can include baselines like PolyGCN [1] and GREET [2].
> > > - Q2. While you mention that your model requires less training time per epoch, you also state that the total number of epochs is larger, making it hard to conclude if the overall training time is better. A clear comparison of total training times would help.
> > > - Main concern. My primary concern is the positioning and practical utility of the proposed approach. While it aims to provide a general graph augmentor that can enhance various graph structures, the results suggest that it does not consistently outperform state-of-the-art baselines in the specific domain. The pre-trained augmentor needs to be further fine-tuned to adapt to each single task. Thus I am curious: under what circumstances is your approach particularly necessary or advantageous?
> > > - It would be highly beneficial to demonstrate whether the proposed augmentor can enhance the performance of SOTA methods when applied to them. Showing a measurable performance improvement when combined with existing SOTA models for each task would provide stronger evidence of the practical value of the approach.
> > >
> > > [1] Chen, Jingyu, et al. PolyGCLL Graph Contrastive Learning via Learnable Spectral Polynomial Filters. ICLR 2024.
> > >
> > > [2] Liu, Yixin, et al. Beyond Smoothing: Unsupervised Graph Representation Learning with Edge Heterophily Discriminating. AAAI 2023.

---

> > > > ### Author Response · Authors · 2024-11-23
> > > > **Reply to reviewer 9F3c**
> > > >
> > > > We thank the reviewer for the follow-up questions and concerns. Below, we summarize the additional content included in this response:
> > > >
> > > > 1. **Node classification results on homophilic graphs**. In Table 6 of the manuscript, we included node classification results on Cora, Citeseer, and Pubmed. These results show that UniAug matches the performance of vanilla GCN and effectively reduces performance discrepancies among nodes with varying homophily ratios.
> > > > 2. **Pre-training time comparison**. We apologize for the ambiguity in the previous table on training time. The total pre-training time is the product of **seconds per epoch** and **number of epochs**. While the total training time for UniAug is longer than that of the baselines, we emphasize that this comparison is conducted under the **multi-domain pre-training setting**, where **most baselines experience a performance decline compared to vanilla GNNs**.
> > > > 3. **Comparison with SOTA**. In Table 4 of the manuscript, we benchmarked UniAug with NCN[1], the current state-of-the-art (SOTA) method for link prediction. Additionally, we performed similar analyses for graph classification using PIN[2], a SOTA method, and node classification using PolyGCL. The results, summarized in the following tables, demonstrate that UniAug consistently outperforms these backbones across tasks, with the sole exception of the Texas dataset.
> > > >
> > > > | Link Prediction | Cora | Citeseer | Pubmed | Power | Yeast | Erdos | Flickr |
> > > > | --- | --- | --- | --- | --- | --- | --- | --- |
> > > > |  | MRR | MRR | MRR | Hits@10 | Hits@10 | Hits@10 | Hits@10 |
> > > > | NCN | 31.72±4.48 | 58.03±3.45 | 38.26±2.56 | 27.36±5.00 | 39.85±5.07 | 36.81±3.29 | 8.33±0.92 |
> > > > | UniAug - NCN | **35.92±7.85** | **61.69±3.21** | **40.30±2.53** | **30.20±1.46** | **42.11±5.74** | **39.26±2.84** | **8.85±0.90** |
> > > >
> > > > | Graph Classification | Proteins | NCI1 | IMDB-B |
> > > > | --- | --- | --- | --- |
> > > > | PIN | 78.8±4.4 | 85.1±1.5 | 76.6±2.9 |
> > > > | UniAug - PIN | **80.2±2.8** | **86.5±1.4** | **77.9±1.8** |
> > > >
> > > > | Node Classification | Cornell | Wisconsin | Texas | Actor |
> > > > | --- | --- | --- | --- | --- |
> > > > | PolyGCL | 82.62±3.11 | 85.50±1.88 | **88.03±1.80** | 41.15±0.88 |
> > > > | UniAug - PolyGCL | **84.31±2.88** | **88.35±2.58** | 86.70±2.77 | **43.01±1.27** |
> > > >
> > > > Based on these results, we highlight the practical utility of UniAug in terms of both performance and flexibility:
> > > >
> > > > 1. **Performance improvements**. As described in the manuscript, UniAug was primarily implemented with vanilla GNNs as backbones (e.g., GCN for node classification and link prediction, and GIN for graph classification). The experimental results demonstrate that **UniAug significantly improves performance compared to vanilla GNNs and outperforms other graph data augmentation methods**. Remarkably, UniAug also outperforms some specifically designed SOTA methods when combined with vanilla GNN backbones. Furthermore, **UniAug continues to enhance performance when integrated with SOTA methods**, setting new SOTAs with the data augmentation paradigm.
> > > > 2. **Plug-and-play flexibility across tasks and backbones**. UniAug is designed to seamlessly integrate with any downstream backbone and consistently improve its performance, as mentioned in the former bullet point. Additionally, UniAug can be combined with other data augmentation methods. For example, as shown in Table 5 of the manuscript, the combination of UniAug and Half-Hop outperforms its counterpart using Half-Hop alone.
> > > >
> > > > [1] Wang, Xiyuan, Haotong Yang, and Muhan Zhang. "Neural Common Neighbor with Completion for Link Prediction." ICLR, 2024.
> > > >
> > > > [2] Truong, Quang, and Peter Chin. "Weisfeiler and lehman go paths: Learning topological features via path complexes." AAAI*,* 2024.

---

> ### Author Response · Authors · 2024-12-02
> **Gentle Reminder**
>
> Thank you for your thoughtful and constructive review. We sincerely appreciate the time and effort you’ve invested in providing such valuable feedback. We would greatly appreciate it if you could share any further questions or comments. We look forward to your reevaluation of our work.
>
> Best regards, The Authors

---

### Author Response · Authors · 2024-11-18
**General response to the reviewers**

We thank the reviewers for their valuable suggestions and concerns. Below, we summarize key experimental results added during the rebuttal period:

1. **Comparison with self-supervised baselines on link prediction and node classification**: To provide a more comprehensive evaluation, we have included both semi-supervised and self-supervised results for graph classification in Appendix D. Additionally, we present self-supervised results for link prediction and node classification in the following tables. The best results are highlighted in bold, with the last column showing the average ranking. Our findings demonstrate that UniAug consistently outperforms the baselines in terms of average ranking, particularly on heterophilic node classification datasets.

|  |  | Cora | Citeseer | Pubmed | Power | Yeast | Erdos | Flickr |  |
| --- | --- | --- | --- | --- | --- | --- | --- | --- | --- |
|  |  | MRR | MRR | MRR | Hits@10 | Hits@10 | Hits@10 | Hits@10 | A.R. |
| Self-supervised | MVGRL[1] | 29.13±3.90 | 51.32±4.12 | 15.21±2.35 | 31.71±3.78 | 23.74±5.74 | 36.21±2.81 | 8.42±2.18 | 4.29 |
|  | GRACE[2] | 31.77±4.31 | 49.13±3.95 | 16.88±1.74 | 28.21±5.04 | 23.96±4.31 | 33.90±2.12 | **9.87±0.98** | 4.00 |
|  | BGRL[3] | 33.59±2.14 | 51.91±5.01 | 16.93±2.03 | 33.71±3.21 | 25.91±3.12 | 37.95±1.73 | 8.52±1.85 | 2.57 |
|  | GraphMAE[4] | 32.98±5.01 | 52.71±5.39 | **18.83±1.30** | 32.81±2.12 | 26.51±2.92 | 35.63±3.61 | 7.01±3.86 | 2.86 |
| GDA | UniAug | **35.36±7.88** | **54.66±4.55** | 17.28±1.89 | **34.36±1.68** | **27.52±4.80** | **39.67±4.51** | 9.46±1.18 | 1.29 |

| ACC ↑ |  | Cornell | Wisconsin | Texas | Actor | Chameleon* | Squirrel* | A.R. |
| --- | --- | --- | --- | --- | --- | --- | --- | --- |
| Self-supervised | MVGRL[1] | 56.19±2.42 | 50.64±5.89 | 61.70±3.94 | 31.37±0.83 | 32.34±2.11 | 35.32±1.32 | 4.00 |
|  | GRACE[2] | 56.39±2.11 | 53.83±3.56 | 63.54±2.57 | 28.14±0.81 | 35.71±1.95 | 33.65±2.51 | 4.17 |
|  | BGRL[3] | 56.67±2.13 | 59.80±4.08 | 65.78±2.66 | 29.80±0.31 | 37.01±2.89 | 34.77±2.01 | 2.67 |
|  | GraphMAE[4] | 57.31±2.11 | 58.27±2.91 | 58.34±3.57 | 28.97±0.27 | 36.75±1.78 | 39.13±2.01 | 3.17 |
| GDA | UniAug | **68.11±6.72** | **69.02±4.96** | **73.51±5.06** | **33.11±1.57** | **43.84±3.39** | **41.90±1.90** | 1.00 |

2. **Data scaling with fixed distribution coverage**. In the manuscript, we conducted a data scaling experiment using three pre-training datasets: SMALL, FULL, and EXTRA. We acknowledge that these datasets differ in both scale and diversity. To isolate the effect of data quantity, we clustered the pre-training set into 10 clusters based on graph-level representations (Section 3.2) and performed stratified sampling within these clusters. Using this approach, we created three subsets containing 25%, 50%, and 75% of the total graphs and pre-trained UniAug on each subset. The results, summarized in the following figures with links for [graph classification](https://drive.google.com/file/d/15wiJj9awBqxYbvJirY99O3RLwZsUR8pv/view?usp=sharing) and [link prediction](https://drive.google.com/file/d/19WicL43j9YEwWHbXZsBI9b93OjOTznJl/view?usp=sharing), reveal a clear trend of performance improvement as the size of the pre-training set increases. Combined with the experiments on the SMALL, FULL, and EXTRA sets, these findings highlight that UniAug benefits from both increasing the scale and enhancing the diversity of the pre-training data.

3. **Few-shot graph classification**: To demonstrate the effectiveness of UniAug in scenarios with limited labeled data, we include results for 5-shot graph classification following [5]. These results show that UniAug achieves significant performance improvements over the self-supervised baselines, underscoring its robustness and adaptability in few-shot settings.

| 5-shot ACC ↑ | Proteins | Enzymes |
| --- | --- | --- |
| GIN | 58.17±8.58 | 20.34±5.01 |
| InfoGraph | 54.12±8.20 | 20.90±3.32 |
| GraphCL | 56.38±7.24 | 28.11±4.00 |
| JOAO | 57.21±6.91 | 35.31±3.79 |
| GraphMAE | 58.03±5.35 | 33.91±6.58 |
| UniAug | **66.85±4.71** | **48.37±4.77** |

[1] Hassani, Kaveh, and Amir Hosein Khasahmadi. "Contrastive multi-view representation learning on graphs." ICML, 2020.

[2] Zhu, Yanqiao, et al. "Deep graph contrastive representation learning." arXiv, 2020.

[3] Thakoor, Shantanu, et al. "Large-Scale Representation Learning on Graphs via Bootstrapping." ICLR, 2022

[4] Hou, Zhenyu, et al. "Graphmae: Self-supervised masked graph autoencoders." KDD, 2022.

[5] Liu, Zemin, et al. "Graphprompt: Unifying pre-training and downstream tasks for graph neural networks." WWW, 2023.

---

### Meta-Review · Area_Chair_mYff · 2024-12-17

**Metareview:**

The UniAug framework proposed in this paper shows some potential in cross-domain graph learning tasks, but several key issues remain. First, there is inconsistency in the comparison with baseline methods, particularly regarding the differences in feature usage, which affects the fairness and comparability of the results. Second, although the proposed method demonstrates some performance improvements in the experiments, its novelty is questioned, as similar data augmentation methods based on diffusion models have already been explored in previous research. While the authors have provided responses, they have not adequately addressed the related concerns. Therefore, I recommend rejecting the submission.

**Additional Comments On Reviewer Discussion:**

While the authors have provided responses, they have not adequately addressed the related concerns.

---

### Decision · Program_Chairs · 2025-01-22

Reject